# iGABASnFR2 is an improved genetically encoded protein sensor of GABA

Ilya Kolb[1,2†], Jeremy P Hasseman[1,2†], Akihiro Matsumoto[3,4], Thomas P Jensen[5], Olga Kopach[5,6], Benjamin J Arthur[2], Yan Zhang[2,7], Arthur Tsang[1,2], Daniel Reep[1,2], Getahun Tsegaye[1,2], Jihong Zheng[1,2], Ronak H Patel[2,7], Loren L Looger[8], Jonathan S Marvin[2,7], Wyatt L Korff[1,2], Dmitri A Rusakov[5], Keisuke Yonehara[4], GENIE Project Team, Glenn C Turner[1,2]*

[1]Janelia GENIE Project Team, Janelia Research Campus, Howard Hughes Medical Institute, Ashburn, United States; [2]Janelia Research Campus, Howard Hughes Medical Institute, Ashburn, United States; [3]Department of Gene Function and Phenomics, National Institute of Genetics, Mishima, Japan; [4]Graduate Institute for Advanced Studies, SOKENDAI, Hayama, Japan; [5]UCL Queen Square Institute of Neurology, University College London, London, United Kingdom; [6]Neuroscience and Cell Biology Research Institute, City St George's University of London Cranmer Terrace, London, United Kingdom; [7]Janelia Tool Translation Team, Janelia Research Campus, Howard Hughes Medical Institute, Ashburn, United States; [8]HHMI, Department of Neurosciences, University of California, San Diego, La Jolla, United States

*For correspondence:
turnerg@janelia.hhmi.org

†These authors contributed equally to this work

Competing interest: The authors declare that no competing interests exist.

## eLife Assessment

This manuscript reports the development and characterization of iGABASnFR2, a genetically encoded GABA sensor that demonstrates substantially improved performance compared to its predecessor, iGABASnFR1. The work is comprehensive and methodologically rigorous, combining high-throughput mutagenesis, functional screening, structural analysis, biophysical characterization, and in vivo validation. The significance of the findings is **fundamental**, and the supporting evidence is **compelling**. iGABASnFR2 represents a notable advance in GABA sensor engineering, enabling enhanced imaging of GABA transmission both in brain slices and in vivo, and constitutes a timely, technically robust addition to the molecular toolkit for neuroscience research.

**Abstract** Monitoring GABAergic inhibition in the nervous system has been enabled by the development of an intensiometric molecular sensor that directly detects GABA. However, the first generation iGABASnFR exhibits low signal-to-noise and suboptimal kinetics, making in vivo experiments challenging. To improve sensor performance, we targeted several sites in the protein for near-saturation mutagenesis and evaluated the resulting sensor variants in a high-throughput screening system using evoked synaptic release in primary cultured neurons. This identified a sensor variant, iGABASnFR2, with 4.1-fold improved sensitivity and 30% faster rise time, and binding affinity that remained in a range sensitive to changes in GABA concentration at synapses. We also identified sensors with an inverted response, decreasing fluorescence intensity upon GABA binding. We termed the best such negative-going sensor iGABASnFR2n, which can be used to corroborate observations with the positive-going sensor. These improvements yielded a qualitative enhancement of in vivo performance when compared directly to the original sensor. iGABASnFR2 enabled the first measurements of direction-selective GABA release in the retina. In vivo imaging in somatosensory cortex revealed that iGABASnFR2 can report volume-transmitted GABA release following whisker

stimulation. Overall, the improved sensitivity and kinetics of iGABASnFR2 make it a more effective tool for imaging GABAergic transmission in intact neural circuits.

## Introduction

Genetically encoded neurotransmitter sensors have significantly advanced neuroscience by enabling direct, real-time monitoring of neurotransmitter dynamics. These tools have facilitated the study of synaptic transmission, input-output relationships within individual neurons, and large-scale network activity. Among these, genetically encoded sensors for GABA are particularly desirable due to the inherent challenges of detecting inhibitory signaling. GABAergic neurotransmission plays a crucial role in shaping neural circuit dynamics and maintaining the balance between excitation and inhibition across the brain. Disruptions in GABAergic signaling have been implicated in neurological and psychiatric disorders, including epilepsy, schizophrenia, and autism, highlighting the importance of developing sensitive tools to study inhibitory transmission (*Sohal and Rubenstein, 2019*).

Although there are other techniques to measure GABA levels in the brain, they are limited in their sensitivity and/or spatiotemporal resolution. Microdialysis allows direct, quantitative measurement of extracellular GABA, even in human patients, but its temporal resolution is restricted to the scale of minutes (*van der Zeyden et al., 2008*). Electrophysiological approaches, while offering high sensitivity and temporal precision, do not scale to large neuronal populations and cannot independently distinguish GABAergic signaling from other hyperpolarizing currents (*Macdonald and Olsen, 1994*). These limitations underscore the need for improved genetically encoded fluorescent sensors capable of resolving GABA dynamics with greater sensitivity, specificity, and temporal resolution.

We previously developed an <u>i</u>ntensity-based <u>GABA Se</u>nsing <u>F</u>luorescent <u>R</u>eporter, iGABASnFR, that increases fluorescence in the presence of GABA (*Marvin et al., 2019*), from here on referred to as iGABASnFR1. Like many sensors (*Marvin et al., 2011*), this design used a periplasmic binding protein from bacteria as the ligand-binding domain, in this case Pf622 from the bacterium *Pseudomonas fluorescens*. A circularly permuted superfolder GFP (cpSFGFP) was then inserted at a site that supports fluorescence changes associated with GABA binding, and the protein was given trafficking sequences to localize it to the membrane. While this sensor enabled optical monitoring of GABAergic activity, its sensitivity and dynamic range remained limited compared to extensively optimized sensors like jGCaMP and iGluSnFR (*Aggarwal et al., 2023*; *Zhang et al., 2023*). In neuronal culture, the original iGABASnFR1 exhibited a half-maximal effective concentration ($EC_{50}$) of ~30 μM and maximal $\Delta F/F$ of ~0.6. A binding pocket mutation (F102G) increased the maximal $\Delta F/F$ of the sensor but at the cost of an increased $EC_{50}$ and poor membrane localization. As such, the overall performance of iGABASnFR1 was not well-suited for high-resolution, photon-limited imaging applications.

To address these limitations, we used site-directed mutagenesis to engineer two next-generation GABA sensors, iGABASnFR2 and iGABASnFR2n. These sensors exhibit improved sensitivities and affinities for GABA, with positive- and negative-going fluorescence responses, respectively. Here, we describe the development and characterization of these enhanced GABA sensors and demonstrate their application in imaging inhibitory neurotransmission.

## Results

### Screening for improved variants

The original iGABASnFR1 (*Figure 1a*, top) was engineered to increase fluorescence in the presence of GABA and express robustly in mammalian neurons, making it a valuable addition to the suite of biosensors available to study the brain (*Dong et al., 2022*; *Looger and Griesbeck, 2012*; *Marvin et al., 2019*; *Yang et al., 2024*). However, iGABASnFR1 is a first-generation sensor, with low sensitivity in vivo, particularly in comparison to other sensors, such as jGCaMP and iGluSnFR, which have been subjected to multiple generations of engineering. Consequently, we applied the same screening pipeline used to optimize jGCaMP (*Dana et al., 2016*; *Wardill et al., 2013*; *Zhang et al., 2023*) to the task of improving iGABASnFR performance. This pipeline uses field stimulation to evoke activity of cultured primary neurons expressing different sensor variants. Stimulation elicits synaptic release, allowing us to screen for improved sensor performance by measuring response amplitude and kinetics (*Figure 1b*), in an experimental setting that approximates in vivo conditions.

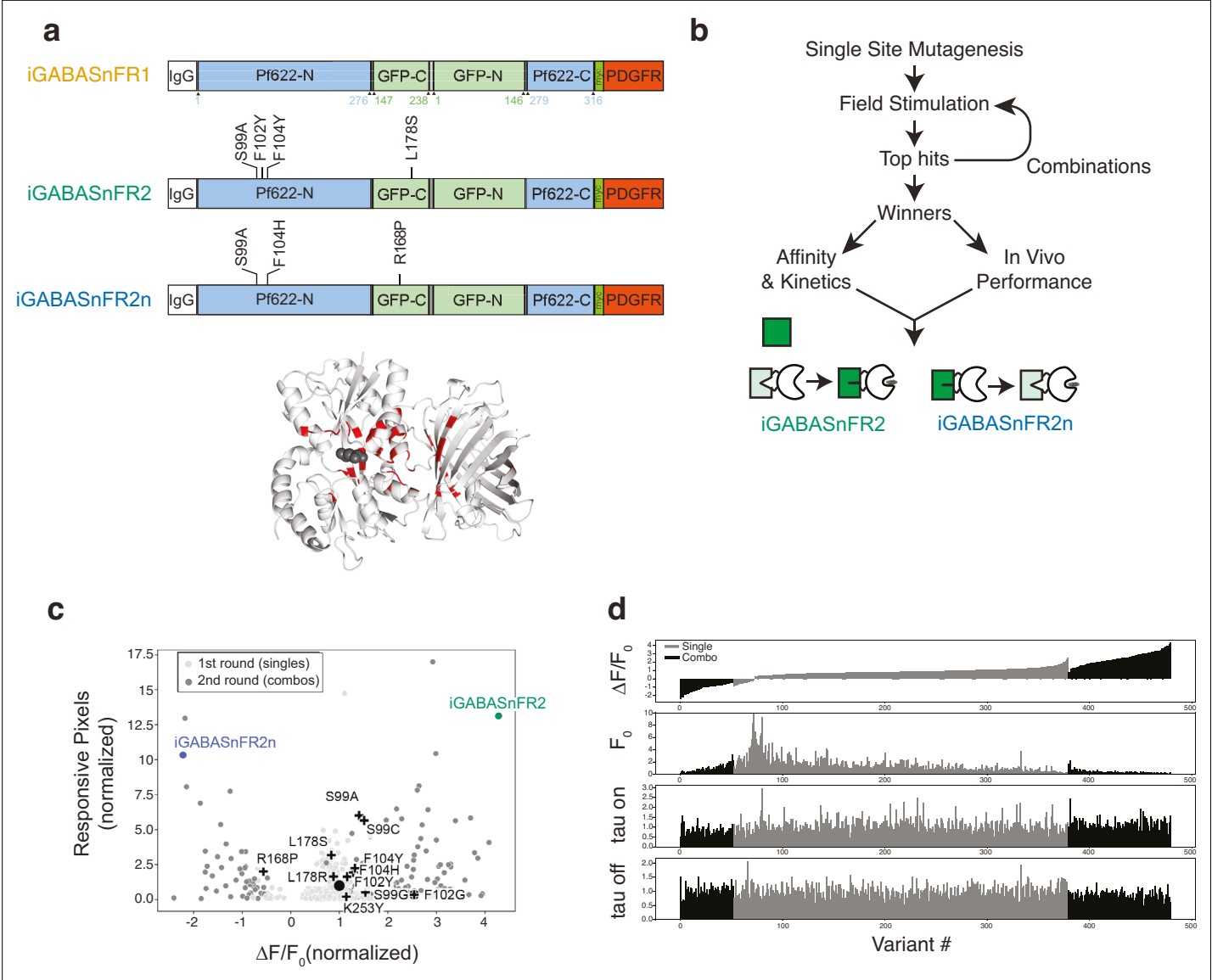

**Figure 1.** Field stimulation screen for improved iGABASnFR. (**a**) Top: Schematic representation of iGABASnFR1 and the amino acid substitutions that gave rise to iGABASnFR2 and iGABASnFR2n. White, IgG secretion signal (cleaved off during trafficking to cell surface); blue, GABA binding protein Pf622; green, cpSFGFP; dark green, Myc epitope tag; red, PDGFR transmembrane domain. Numbering is relative to each of the constituent protein domains; the relationship to sequential numbering of the entire polypeptide is shown in *Figure 1—figure supplement 1*. Bottom: Crystal structure of a preliminary version of iGABASnFR (PDB ID 6DGV) with the 39 sites targeted for mutagenesis. Gray spheres indicate the approximate position of GABA, based on homology to the liganded structure of GABA-binding protein Atu4243 (*Planamente et al., 2012*). (**b**) Mutagenesis and screening strategy. Variants with single-site mutations are screened for ΔF/F and expression in an initial field stimulation assay, strong performers identified and then combined and re-screened in a second round. The top-performing mutants (positive-going quadruple mutant iGABASnFR2 and negative-going triple mutant iGABASnFR2n) are characterized further. (**c**) Joint optimization of sensitivity (ΔF/F, x-axis) and expression, measured with responsive pixels (y-axis). ΔF/F is normalized to in-plate iGABASnFR1 controls. Black circle: (1,1) position represents iGABASnFR. Plus signs: mutants from the first round that were put together to form combos in the second round. iGABASnFR2 and 2 n exhibited increased ΔF/F (4.3-fold and –2.2-fold of iGABASnFR, respectively) and greater numbers of responsive pixels (13.1-fold and 10.3-fold). Illumination: 0.34 mW/mm², imaging framerate: 50 Hz. (**d**) Performance measures of sensor variants from the screen. Single-site variants are shown in gray, and double-site combinations are shown in black. Variants are ranked according to the $\Delta F/F_0$ values measured for 40 AP, and those rankings are maintained for the lower displays of sensor $F_0$, tau on, the time constant of the rising phase of the response, and tau off for the decay. All values are normalized to in-plate iGABASnFR1 controls.

The online version of this article includes the following figure supplement(s) for figure 1:

**Figure supplement 1.** Annotated amino acid sequence of iGABASnFRs.

Sensor variants were derived by near-saturation mutagenesis of selected sites within the protein, and performance was evaluated in a first round of screening. To select improved variants, we jointly optimized for both the ΔF/F of the sensor and its expression levels. Screening for expression was necessary because, in our previous effort to engineer iGABASnFR (*Marvin et al., 2019*), we discovered several mutations, most notably F102G, that increased the maximal ΔF/F but exhibited expression problems in neurons, with low baseline fluorescence and intracellular aggregates. We quantified the expression levels of different variants by counting the total number of pixels in a well that showed a significant increase in fluorescence during the field stimulation period. These pixels were classified as responsive pixels.

To guide our site-selection efforts, we used the unliganded crystal structure of a preliminary version of iGABASnFR https://www.rcsb.org/structure/6dgv ; (*Marvin et al., 2019*), a strategy that worked well for optimizing GCaMP (*Akerboom et al., 2009*). Specifically, we identified 39 sites to target: 14 near the GABA binding site, 6 at or around the protein hinge area, 10 on the cpSFGFP, and 9 at the interface between the two ligand-binding and cpGFP domains (*Figure 1a*, bottom).

In the first round of screening, these 39 sites were targeted for mutagenesis, generating 9±5 substitutions at each site (range: 1–19) for a total of 3947 variants. Sensor variants were arrayed in 96-well plates, with each plate containing four replicate wells of the same variant and eight replicate wells of the original iGABASnFR, which served as part of our quality control measures. Each well received 1, 10, and 40 field stimulation pulses separated by 12 ms, and we calculated responses by averaging fluorescence changes over all the responsive pixels in each well. We found 93 mutants with ΔF/F significantly higher than in-plate iGABASnFR1 controls, and 22 of those also had a higher fraction of responsive pixels than iGABASnFR, indicating improved expression (*Figure 1c*). Interestingly, at least 20 sequence-confirmed mutations, located in 5 different positions - $R168_{gfp}$, $D276_{Pf}$, $T203_{gfp}$, $V150_{gfp}$, and $V272_{Pf}$ - inverted the response of the sensor, making it decrease in fluorescence with field stimulation (number refers to location in the protein domain denoted by subscript, either cpSFGFP or the ligand binding Pf622). However, none of the inverted sensors exhibited |ΔF/F| greater than that of iGABASnFR. Among them, the $R168_{gfp}P$ mutant was the most promising, with relatively high ΔF/F (0.6x of iGABASnFR) and expression (2x responsive pixels of iGABASnFR; *Figure 1c*).

Although we found many marginally improved variants in the first round of screening, there was no clear winner that could be considered to be a major improvement over iGABASnFR. The most sensitive positive-going mutant in the screen was the previously known mutation $F102_{Pf}G$ (ΔF/F 2.5x of iGABASnFR) which was confirmed to have poor expression (0.3x responsive pixels of iGABASnFR). Similarly, the top-expressing mutant in the screen ($L178_{gfp}Y$; 14.8x responsive pixels of iGABASnFR) had only a modest ΔF/F (1.1x of iGABASnFR). Therefore, in the first round of screening, we succeeded in improving ΔF/F and expression separately but not jointly.

## Enhancing performance with combinations of single-site mutants

We hypothesized that we could generate variants with both improved sensitivity and expression by combining mutations that conveyed each property separately. To test this, the mutations $S99_{Pf}\rightarrow A/G/C$, $F102_{Pf}\rightarrow G/Y$, $F104_{Pf}\rightarrow Y/H$, $R168_{gfp}\rightarrow P$, $L178_{gfp}\rightarrow R/S$, and $K253_{Pf}\rightarrow I/Y$ (labeled in *Figure 1d* as plus signs) were chosen based on their performance in the first round and combined to create 635 double mutants, which were then screened in a second round. A large percentage of these mutants (49%) did not pass quality control to be considered for further analysis. The majority of these mutants had poor expression, no detectable response to field stimulation, or both. Of the remaining mutants, 52 exhibited improved expression and ΔF/F over iGABASnFR. More importantly, 27 mutants exhibited ΔF/F higher than $F102_{Pf}G$ and expression better than iGABASnFR, suggesting that our hypothesis was correct and the beneficial properties of the mutants could be additive. Interestingly, many of the mutations that improved ΔF/F and expression in positive-going sensors did the same in negative-going sensors.

Of these variants, the best combination of dynamic range and expression was the mutant iGAB-ASnFR.$S99_{Pf}A.F102_{Pf}Y.F104_{Pf}Y.L178_{gfp}S$ (hereafter iGABASnFR2). The best variant with inverted signal, iGABASnFR.$S99_{Pf}A.F104_{Pf}H.R168_{gfp}P$ was designated iGABASnFR2n (for <u>n</u>egative-going). When we quantified rise and decay time constants, peak ΔF/F, and signal-to-noise, iGABASnFR2 performance was broadly superior to the original across all field stimulation conditions (*Figure 2*). For 10 action potentials (APs), iGABASnFR2 exhibited a peak ΔF/F 4.1-fold greater than iGABASnFR1 (iGABASnFR2

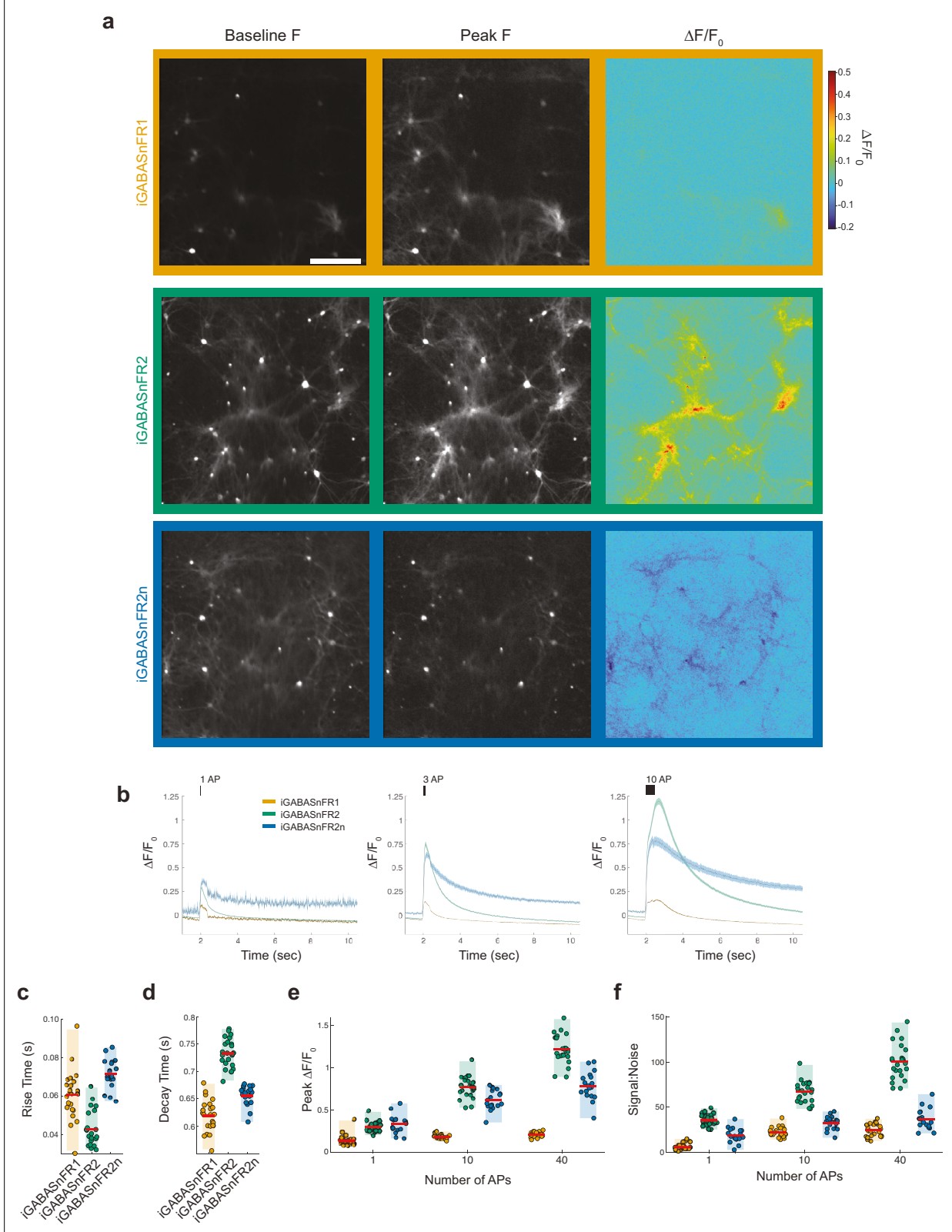

**Figure 2.** Characterization of iGABASnFR variants in cultured neurons. (**a**) Fluorescence images of primary neurons expressing iGABASnFR1 (orange), iGABASnFR2 (green), iGABASnFR2n (blue) under the CAG promoter at baseline, at peak brightness after field stimulation with 40 action potentials (APs), and the corresponding ΔF/F0. Scale bar, 20 μm. (**b**) Time courses of the ΔF/F0 response to 1, 10, and 40 APs delivered at 83 Hz. iGABASnFR2n signals are inverted for display. Traces and error bars denote mean ± s.e.m., n=20 culture wells for each variant. (**c**) Rise time constants of the three

*Figure 2 continued on next page*

Figure 2 continued

sensor variants obtained from exponential fits for the 1 AP condition (n=24 culture wells for iGABASnFR1 and iGABASnFR2, n=18 for iGABASnFR2n). For panels (**c-f**), red lines indicate the mean and boxes indicate the 95% confidence interval. (**d**) Decay time constants of the three variants for the 1 AP condition. (**e**) Peak ΔF/F0 of the three variants over different levels of stimulation. (**f**) Signal-to-noise (d') of the three variants over different levels of stimulation.

0.77±0.13; iGABASnFR1 0.19±0.02; $p<0.001$ Tukey's HSD post hoc test following one-way ANOVA), and a signal-to-noise ratio (SNR) threefold higher (iGABASnFR2 67.6±11.9; iGABASnFR1 22.7±5.5; $p<0.001$). Rise time constants were slightly faster (iGABASnFR2 43±9 ms; iGABASnFR1 61±13 ms; $p<0.001$), while decay times were slower (iGABASnFR2 73±26 ms; iGABASnFR1 62±29 ms; $p<0.001$). The negative-going iGABASnFR2n exhibited slightly reduced performance relative to iGABASnFR2 but still achieved a 3.3-fold greater peak ΔF/F (iGABASnFR2n 0.62±0.12; iGABASnFR1 0.19±0.02; $p<0.001$), and 40% higher SNR compared to iGABASnFR1 (iGABASnFR2n 32.5±7.5; iGABASnFR1 22.7±5.5; $p<0.01$). However, its kinetics were slower, with a rise time of 72±8 ms. These measures clearly indicate that this next generation of GABASnFRs exhibits broadly improved performance in detecting GABA dynamics during synaptic release.

## Structure of the iGABASnFR2-GABA complex

To gain some insight into the structure-function relationship of this sensor, we solved the crystal structure of iGABASnFR2 in complex with GABA (*Figure 3*, *Figure 3—source data 1*; PDB ID 9D57). To examine the conformational changes that accompany GABA binding, we compared this new liganded structure with the apo form of a precursor of the original iGABASnFR (PDB ID 6DGV). Superimposing the structures, using cpGFP as a reference, revealed that the two lobes of the Venus flytrap domain in Pf622 shift closer together to secure GABA in the binding pocket (*Figure 3a*). This conformational change resembles those seen in other bacterial periplasmic amino acid-binding proteins (*Quiocho and Ledvina, 1996*).

The hinge region of Pf622, which connects the two lobes, likely plays a key role in allosteric modulation of ligand binding (*Marvin and Hellinga, 2001*; *Telmer and Shilton, 2003*); (*Marvin et al., 2019*; *Planamente et al., 2012*; *Planamente et al., 2010*). Within that hinge region, the F101L mutation, introduced in the original iGABASnFR1 (*Marvin et al., 2019*), increased binding affinity 10-fold (*Figure 3b*). Also in the hinge region, residue S99 forms hydrogen bonds with the OD2 atom of D59 and NE2 atom of Q17, while the S99A mutation in iGABASnFR2 abolishes these interactions, potentially making the hinge region more flexible.

The GABA binding site itself is defined by the side chains of W9, T13, F100, Y102, W202, R205, D228, and Y264 (*Figure 3c*). In iGABASnFR2, two mutations near the ligand binding site enhance interactions: F102Y forms a hydrogen bond via its hydroxyl group with D228, which, in turn, makes a hydrogen bond with the GABA amino group. And F104Y, while farther from the binding site, helps position W202 to interact with GABA through van der Waals forces. These likely contribute to the increased affinity for GABA we describe below.

Mutations at the interface between the ligand-binding domain and cpGFP modulate ligand-binding-induced fluorescence changes in many sensors (*Akerboom et al., 2009*; *Ding et al., 2014*; *Zhang et al., 2023*). In iGABASnFR2, the L178$_{gfp}$S mutation at this interface is likely involved in a hydrogen-bonding network with nearby hydrophilic residues, enhancing the GABA-induced fluorescence change (*Figure 3b*). The N260A mutation in iGABASnFR1 eliminates side chain hydrogen bonding, removes a cryptic N-linked glycosylation site (N-X-S/T), and enhances protein expression. Another critical mutation, F145$_{gfp}$W, introduced in the original iGABASnFR1 to improve ΔF/F0, interacts directly with the hydroxyl group in the chromophore and surrounding residues.

Interestingly, although the Venus flytrap domain undergoes significant conformational changes upon GABA binding, cpGFP and its flanking linkers (along with up to four adjacent residues) do not show notable conformational changes (*Figure 3a*). The structural difference between the unliganded and liganded forms shows a root mean square deviation of only 0.25 Å. This finding contrasts with observations in GCaMP (*Akerboom et al., 2012*; *Akerboom et al., 2009*; *Zhang et al., 2023*), where calcium binding to calmodulin induces substantial conformational changes at the interface, contributing to the large change in fluorescence with this sensor. These differences suggest a potential strategy to further enhance the performance of iGABASnFR2.

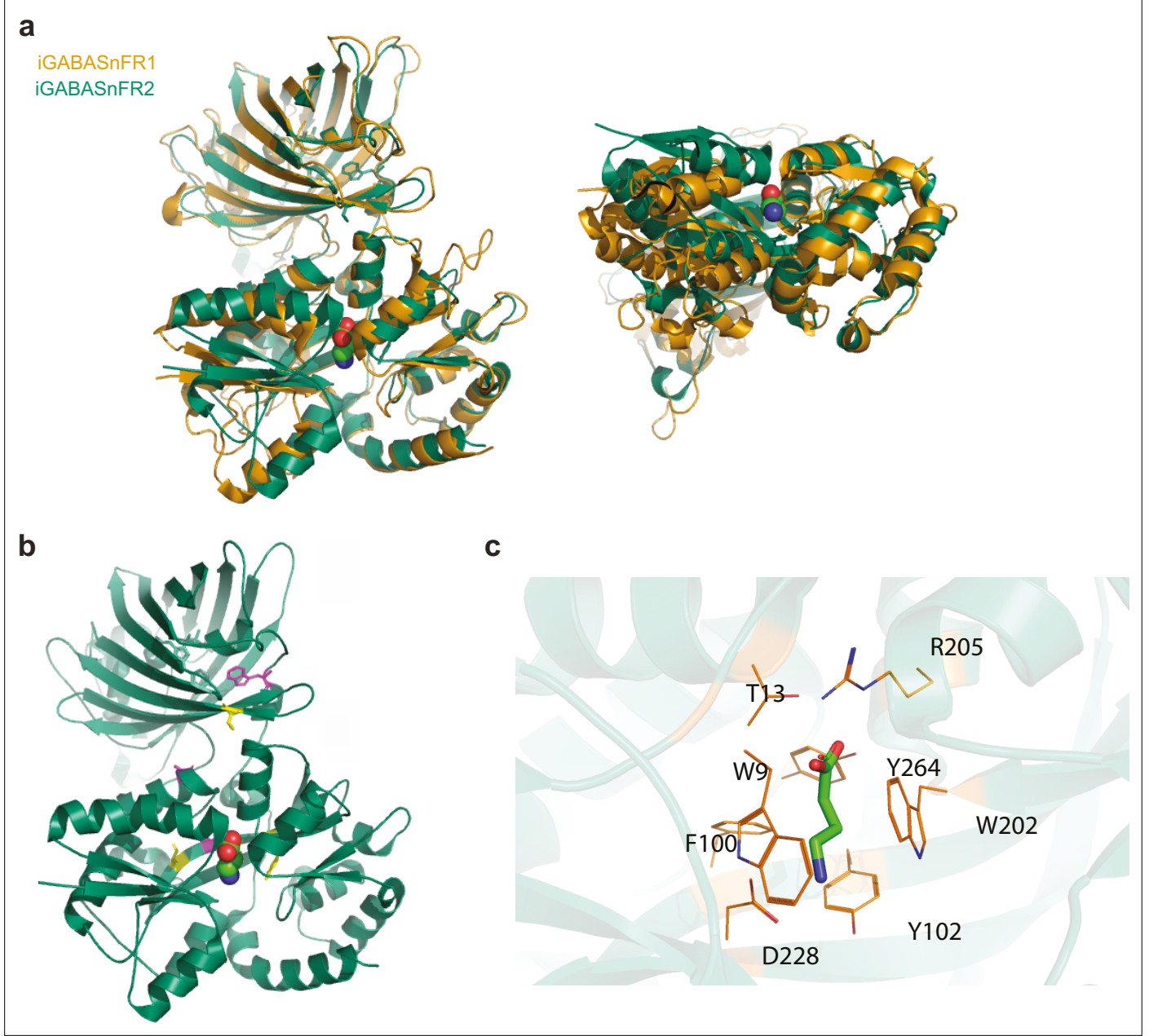

**Figure 3.** Crystal structure of iGABASnFR2 in complex with GABA. (**a**) Superposition of the structure of iGABASnFR2 in complex with GABA (green; PDB ID: 9D57) and unliganded iGABASnFR precursor (orange). The bound GABA molecule is represented as colored spheres. (**b**) Cartoon representation of iGABASnFR2 with key mutations shown as yellow sticks. Mutations in the original iGABASnFR1 are shown in magenta. (**c**) GABA binding site of iGABASnFR2. Residues involved in GABA binding are shown as lines and GABA is shown as sticks.

The online version of this article includes the following source data for figure 3:

**Source data 1.** Data collection and refinement statistics of iGABASnFR2 in complex with GABA.

## Biochemical characterization

For in vivo applications, the effectiveness of the sensor depends on both its affinity for GABA, as well as the kinetics of GABA binding. Sensor affinity should not be so high that it is saturated by tonic levels of GABA in the brain, but should of course be in a range sensitive to the changes in concentration achieved during synaptic release. In mammals, extrasynaptic GABA concentrations lie in the low micromolar range 0.2–2.5 µM (*Glykys and Mody, 2007*; *Lerma et al., 1986*; *Roth and Draguhn, 2012*; *Tossman et al., 1986*), while synaptically released GABA can transiently reach low

millimolar concentrations (~1.5–3 mM *Barberis et al., 2004*; *Mozrzymas et al., 2003*; *Roth and Draguhn, 2012*). The half-maximal effective concentration (EC$_{50}$) of iGABASnFR1 when it is expressed on the surface of neurons is 30 µM, so increasing sensor affinity could potentially lead to improved performance in vivo without contaminating signal from tonic GABA levels. However, kinetics are also important - the rapid changes in GABA concentration during synaptic release mean that sensor kinetics have to be fast enough to detect the change before GABA concentrations drop back down due to reuptake and diffusion.

We first determined sensor affinities by titrating GABA concentration while monitoring fluorescence of purified protein. Somewhat surprisingly, purified iGABASnFR2 showed a smaller dynamic range than iGABASnFR1, although its affinity was higher (*Figure 4a*; max dF/F0 iGABASnFR2: 0.45, iGABASnFR1: 1.82; EC$_{50}$ iGABASnFR2=1.1 µM, iGABASnFR1=5.7 µM). It also displayed high selectivity for GABA over structurally related compounds (*Figure 4—figure supplement 1*), none of which interfered with GABA binding (*Figure 4—figure supplement 2*). In the experiments shown in *Figure 4—figure supplement 2*, the apparent EC50 of iGABASnFR2 was similar to that measured under other conditions, although the shape of the dose-response curve differed from that observed in other assays for reasons that are not currently clear. Importantly, despite this difference, none of the tested compounds acted as strong non-competitive allosteric antagonists or inhibitors of GABA binding. Previous work with iGluSnFR has shown that titrations with purified protein can yield different affinity values than when expressed on the membrane of cultured neurons (*Aggarwal et al., 2023*). Although the origins of this discrepancy remain unclear, on-cell titrations more closely reflect the operating environment of the sensor and are, therefore, likely to better predict in vivo performance. We found that on cells, the half-maximal effective concentration (EC$_{50}$) of iGABASnFR2 was 6.4±0.21 µM, a sevenfold higher affinity than iGABASnFR1 and 22-fold higher than iGABASnFR. F102$_{Pf}$G (*Figure 4b*). So the improved performance we observed likely stems, in part, from this increased on-cell affinity, which nevertheless remains above background levels of GABA measured in the mammalian brain.

We examined sensor kinetics to step changes in GABA concentration using stopped-flow measurements. Relative to iGABASnFR1, the observed reaction rate constants were far greater for both iGABASnFR2 and iGABASnFR2n (*Figure 4c*). iGABASnFR1 exhibits biphasic kinetics, with a relatively fast initial change compounded with a much longer phase before sensor saturation is reached (*Marvin et al., 2019*). In contrast, both iGABASnFR2 and iGABASnFR2n kinetics were accurately captured by fitting with a single exponential function, indicating a less complex relationship between ligand binding and changes in fluorescence (*Figure 4d*). Although significantly faster than the first version, the kinetics of iGABASnFR2 and 2n are still slower than iGluSnFR3 (*Aggarwal et al., 2023*), suggesting further improvements in sensor performance would be possible if kinetics could be accelerated.

The 1p and 2p spectra of all sensors were similar (*Figure 4e and f*), as were their apparent pKa values (*Figure 4—figure supplement 3*). Notably, iGABASnFR2 and 2n responses were less pH-dependent than those of the first-generation sensor. Additional biophysical properties are reported in *Table 1*. Overall, both kinetic and thermodynamic observations suggest that iGABASnFR2 and 2n offer a major improvement in sensing GABA over the previously available sensor. We next tested this by evaluating performance in vivo.

## Evaluating sensor performance in intact retina

To evaluate indicator performance in an intact neural circuit, we examined synaptic transmission in the retina. GABAergic inhibition is believed to play a pivotal role in generating direction-selective responses to motion in the retina (*Briggman et al., 2011*; *Yonehara et al., 2011*). Direction selectivity is thought to originate in starburst amacrine cells (SACs), which extend radially symmetric dendrites from a centrally located soma. Inputs to SACs are sprinkled across the entire dendritic arbor, while the outputs are confined to varicosities located near the dendritic tips.

Motion selectivity is proposed to arise from differential GABA release. Specifically, the hypothesis is that GABA release is stronger when a visual stimulus moves so that excitation sweeps down the dendritic branch from root to tip (centrifugal motion) rather than when it sweeps from tip to root (centripetal motion) (*Figure 5a*). The resulting direction-selective inhibition is relayed to direction-selective retinal ganglion cells (DSGCs) through asymmetric synaptic connections (*Briggman et al., 2011*; *Yonehara et al., 2011*; *Figure 5b*). By inhibiting DSGC responses to motion in the null direction,

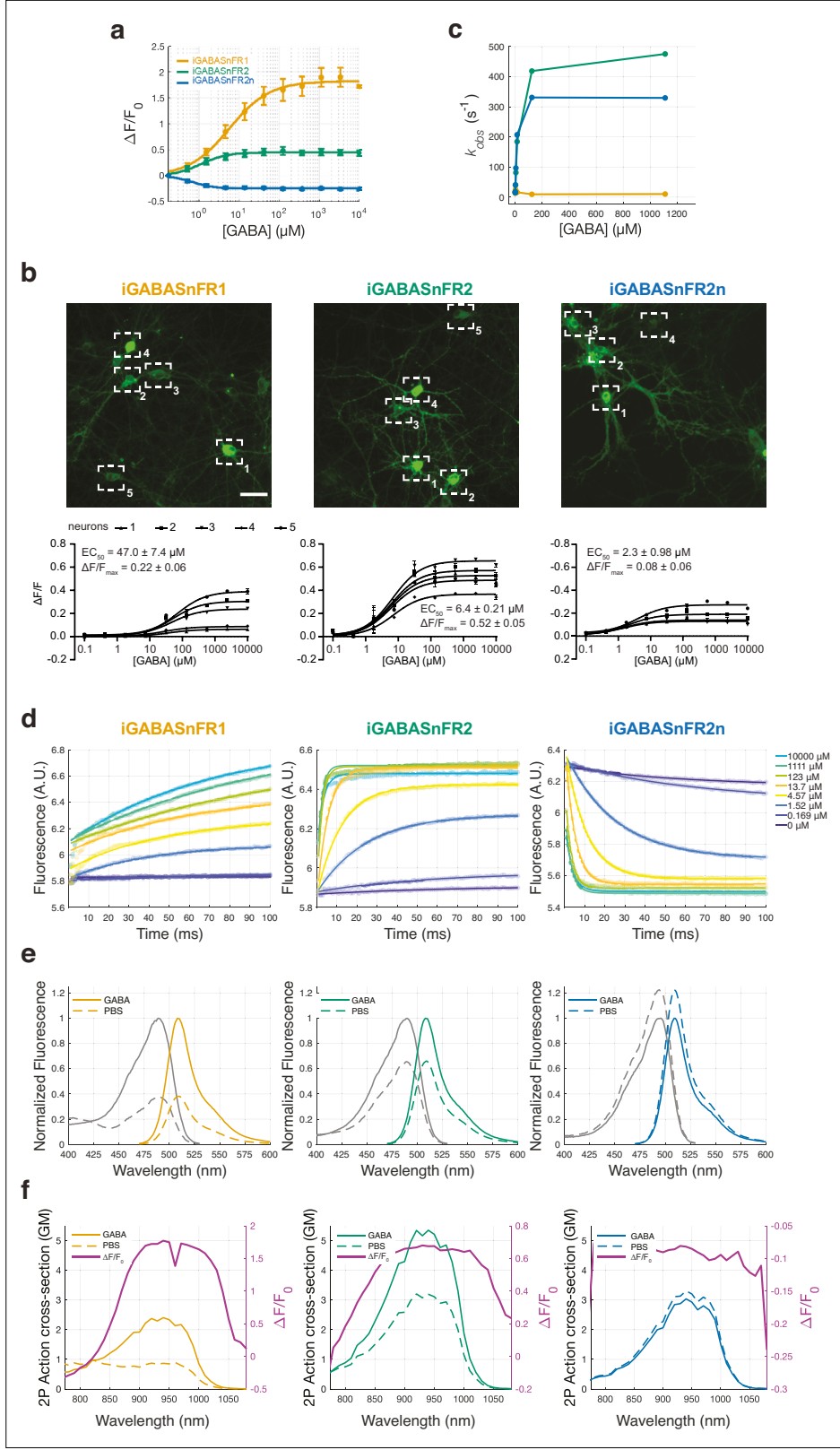

**Figure 4.** Biophysical properties of iGABASnFR variants. (**a**) GABA titrations with purified iGABASnFR protein. Lines indicate fits to mean of n=5 titration series, error bars are s.e.m. (**b**) GABA titrations with sensors expressed on the surface of cultured neurons. Fits (bottom panel) show $\Delta F/F_0$ response to increasing concentrations of GABA measured with region of interests (ROIs) placed on individual cell bodies (top panel). In these conditions,

*Figure 4 continued on next page*

*Figure 4 continued*

iGABASnFR2 shows a greater dynamic range than iGABASnFR1, in contrast to results with purified protein. Color lookup table is the same for all images. Scale bar: 50 µm. Illumination: 5.6 mW/mm$^2$, imaging at 1 frame per second. (**c**) Observed reaction rate constant ($K_{obs}$) values from stopped-flow measurements for the three sensors. (**d**) Stopped-flow kinetics of iGABASnFR variants. Lines indicate fits to mean of n=3 replicates from three separate batches of purified protein. (**e**) One-photon excitation and emission spectra of soluble iGABASnFR protein in the presence (10 mM) and absence of GABA. (**f**) Two-photon excitation spectra and the computed ΔF/F0 of iGABASnFR variants in the presence (10 mM) and absence of GABA.

The online version of this article includes the following figure supplement(s) for figure 4:

**Figure supplement 1.** Responses of iGABASnFR variants to GABA-related compounds.

**Figure supplement 2.** Competition of GABA-related compounds for binding to iGABASnFR variants.

**Figure supplement 3.** pH titrations of iGABASnFR variants.

---

SAC inputs ensure that DSGC output is highly selective for motion in the preferred direction, which is subsequently transmitted to the brain.

Notably, the evidence for this centrifugal selectivity has been obtained by electrophysiological membrane potential recording and two-photon Ca$^{2+}$ imaging from starburst cells (*Euler et al., 2002*; *Vaney et al., 2012*). Direct evidence for direction-selective GABA release has been lacking, and imaging GABA release from the starburst cell dendrites would provide important confirmation of the origin of retinal direction selectivity.

We virally transfected starburst amacrine cells with iGABASnFR1 or iGABASnFR2, using a Cre-Lox system for cellular specificity. Six to eight weeks after injection, we dissected retinae and performed two-photon imaging, analyzing activity within small fields of view expected to contain multiple release sites on the amacrine cell dendrites.

We first tested whether we could detect responses to light flashes with a 500 µm spot that covered the entire field of view. iGABASnFR1 showed weak but measurable signals to this full-field stimulation, although detecting responses on single trials was challenging. However, when presented with moving dots, the weak signal made it very difficult to detect direction selectivity, even after trial-averaging. By contrast, iGABASnFR2 responses to static flashes were large enough to be detected on single trials, and robust direction selectivity of GABA release was readily observed in response to motion stimuli.

We quantified the quality of the signals by computing two measures: (1) response amplitude index (RAI), which denotes the light-evoked response strength and (2) response reliability, which reflects trial-to-trial variance (Methods) (*Baden et al., 2016*). Overall, both the response amplitudes and the reliability of the responses were significantly higher with iGABASnFR2 than with iGABASnFR1 (*Figure 5i and j*). In particular, response reliability was much improved in iGABASnFR2 (mean ± SD iGABASnFR2: 0.66±0.14; iGABASnFR: 0.41±0.11), indicating that this sensor reveals activity that previously would have been below detection threshold.

We also examined the signal-to-noise ratio (SNR) of the responses to motion, comparing response amplitude to the variance of signals at baseline. Compared with iGABASnFR1 signals, iGABASnFR2 signals had significantly higher SNR with both larger signals and lower variance of the baseline (*Figure 5k and l*). The improved SNR and higher response reliability across trials resulted in stable motion responses with lower circular variance (CV; *Figure 5m*), yielding better direction selectivity measurements overall. Overall, the improved performance of iGABASnFR2 not only enabled us to detect more responses but also provided a more accurate measure of GABA release evoked by visual motion, directly demonstrating that starburst cells release GABA in a direction-selective manner.

**Table 1.** Photophysical properties of iGABASnFR variants as purified proteins.

| | $\lambda_{abs}$ (nm) | $\lambda_{Ex}$ (nm) | $\lambda_{Em}$ (nm) | ΔF/F | ε (M/cm$^2$) | | Φ | | τ (ns) | |
|---|---|---|---|---|---|---|---|---|---|---|
| | | | | | GABA | PBS | GABA | PBS | GABA | PBS |
| iGABASnFR1 | 490 | 489 | 508 | 1.9 | 14,070±600 | 5375±220 | 0.58±0.01 | 0.54±0.03 | 2.22±0.05 | 2.32±0.01 |
| iGABASnFR2 | 490 | 490 | 508 | 0.46 | 26,195±390 | 19,800±380 | 0.61±0.02 | 0.54±0.01 | 2.29±0.08 | 2.30±0.02 |
| iGABASnFR2n | 493 | 496 | 509 | −0.13 | 12,390±690 | 13,000±220 | 0.72±0.02 | 0.76±0.01 | 2.71±0.02 | 2.72±0.02 |

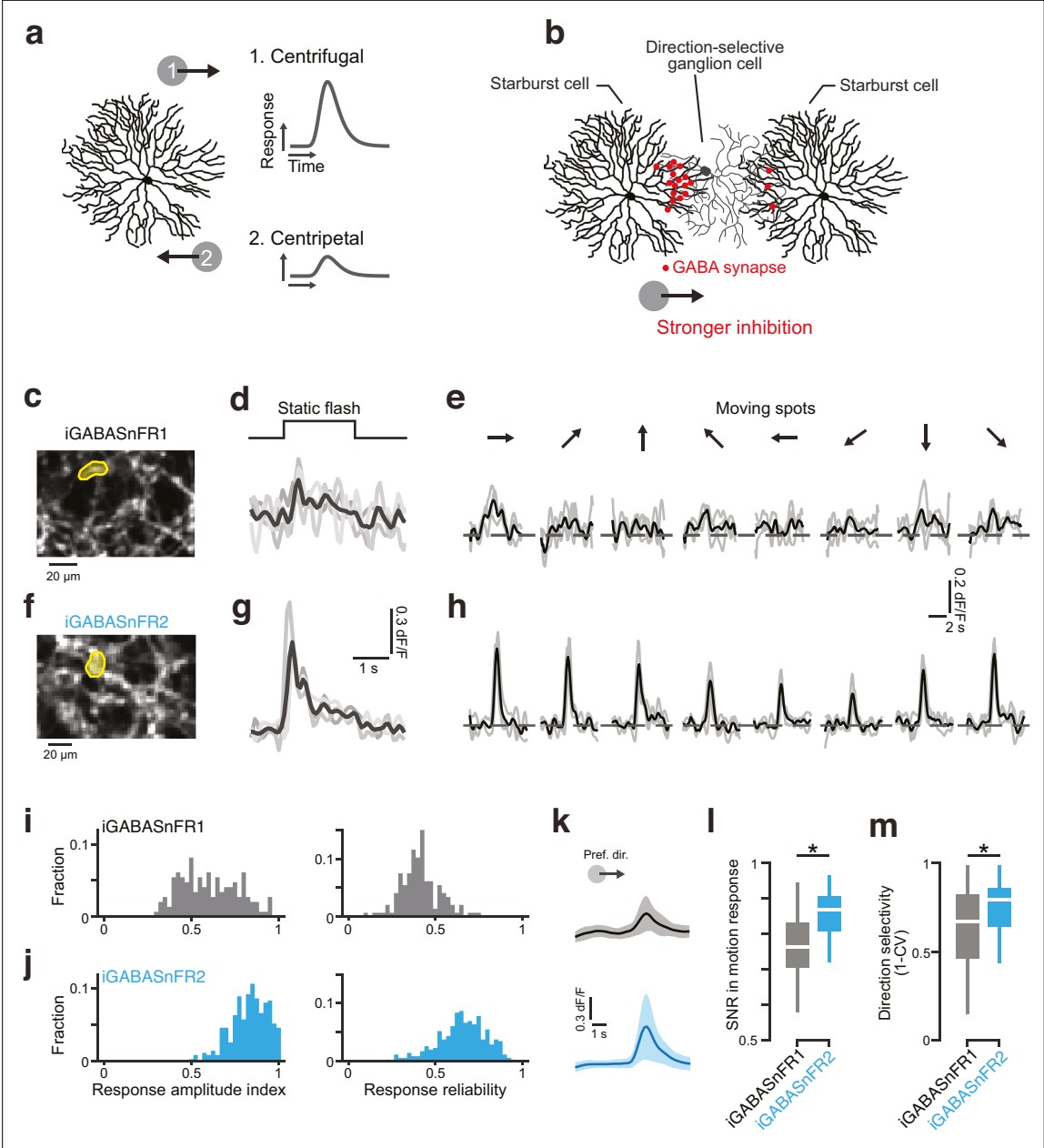

**Figure 5.** iGABASnFR2 reliably reports direction selectivity in the retina. (**a**) Schematic illustration of centrifugal direction selectivity in starburst cell motion responses. (**b**) Spatially asymmetric inhibitory connections (red dots) between starburst cells (SACs) and direction-selective ganglion cells (DSGCs), which are proposed to generate direction selectivity in DSGCs. (**c**) Example field of view of SAC processes expressing iGABASnFR1. The yellow region of interest (ROI) is analyzed to evaluate responses to visual stimuli. (**d**) Responses to static flash from ROI in **c** but with SACs expressing iGABASnFR2. (**e**) Responses to motion stimulus. (**f–h**) Results of imaging using iGABASnFR2. (**i**) Histograms of response amplitude index (left) and response reliability (right) from SACs expressing iGABASnFR. n=147 ROIs collected across five retinae. (**j**) As in **i** but with SACs expressing iGABASnFR2. n=346 ROIs collected from three retinae. Responses to motion stimulus. (**k**) Average signals during preferred direction motion with iGABASnFR1 (top, gray) and iGABASnFR2 (bottom, cyan). Line and shading indicate mean ± s.d (n=147 for iGABASnFR, n=346 for iGABASnFR2). (**l**) Comparison of signal-to-noise ratio (SNR) of the motion response detected with the two sensor versions. $p=0$. Two-tailed Mann-Whitney-Wilcoxon test. (**m**) Comparison of direction selectivity (CV, circular variance) for the two sensor versions. $p=7.812×10^{-6}$. Two-tailed Mann-Whitney-Wilcoxon test.

## Sensor performance at individual cell axons and in vivo

As a first step toward in vivo application, we tested whether iGABASnFR could detect GABA release from single interneuron activation in brain slices. We expressed either iGABASnFR1 or iGABASnFR2 in hippocampal neurons, using viral delivery in mice to obtain acute slices (viral titre 0.1E10), or by

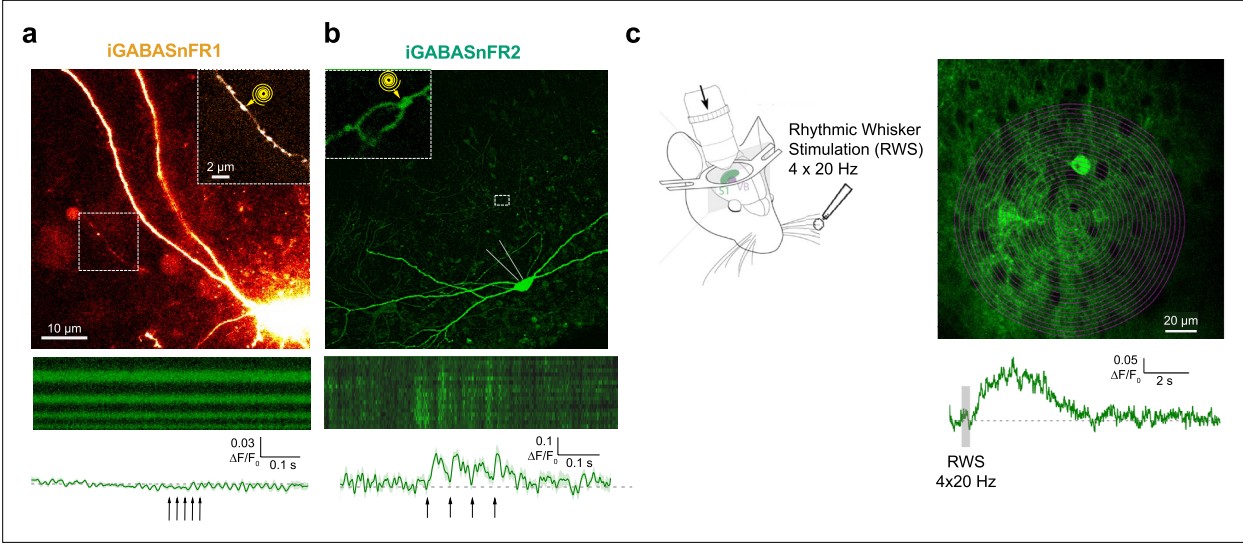

**Figure 6.** iGABASnFR2 reliably detects synaptic GABA release in slices and sensory-evoked GABA in vivo. (**a**) *Top*: Image from a whole-cell recording of an iGABASnFR1-expressing hippocampal interneuron in area CA3 of an acute brain slice. To identify axonal boutons, morphology was visualized using Alexa Fluor 594 (red channel), included in the internal solution. The image shown is from the red channel. The axonal segment (dotted rectangle) is shown magnified in the inset, with the position of a schematized 1.5 µm-wide Tornado scan path indicated. The image is the average projection of a z-stack covering 50 µm. *Bottom*: iGABASnFR1 fluorescence signal acquired using a 0.5 kHz Tornado scan at the bouton shown above. Horizontal axis: time (700 ms scan duration); vertical axis: spiral turn angle. Trace shows mean $\Delta F/F_0$ response (± SEM) across 15 trials of five action potentials at 50 Hz (arrows). (**b**) *Top*: As in (**a**), except the image shows iGABASnFR2 fluorescence from an interneuron in area CA1 in an organotypic slice. The image is the average projection of a 30 µm z-stack. *Bottom*: As in (**a**), but showing iGABASnFR2 signal from the Tornado scan. Trace shows the mean $\Delta F/F_0$ response (± SEM) across 11 trials of four action potentials delivered at 20 Hz (arrows). (**c**) Brief rhythmic whisker stimulus triggers volume-transmitted extracellular elevations of GABA in the barrel cortex detected by iGABASnFR2 fluorescence. *Left*: Schematic of experimental arrangement, with whisker stimulation via four air puffs at 20 Hz while imaging the contralateral barrel cortex. *Right*: The image panel shows the barrel cortex area (~300 µm depth) and position of a 1 kHz Tornado scan. Trace is the single-trial fluorescence response of iGABASnFR2 to a contralateral 200 ms whisker stimulation, as indicated.

The online version of this article includes the following video for figure 6:

**Figure 6—video 1.** iGABASnFR2 detects volume-transmitted GABA signals evoked by sensory stimulation in vivo in the barrel cortex.
https://elifesciences.org/articles/108319/figures#fig6video1

biolistic transfection in organotypic hippocampal slice cultures (see Methods). We performed simultaneous two-photon imaging and whole-cell recordings of individual interneurons, using an intracellular dye to trace their axons and identify presumptive presynaptic boutons. Once individual boutons were identified, we triggered brief bursts of action potentials and monitored iGABASnFR signals using a rapid spiral ('Tornado') scanning approach previously developed for imaging glutamate sensors (*Jensen et al., 2019*). In interneurons expressing iGABASnFR1, we were unable to detect any spike-evoked fluorescence signals, despite conducting 15 trials per cell across five separate experiments (example in *Figure 6a*). In contrast, iGABASnFR2-expressing cells consistently displayed evoked signals at individual axonal boutons (although obtaining a signal-to-noise ratio >3 might require 5–10 trial averaging; example in *Figure 6b*).

We next sought to detect GABA release signals in vivo. In vivo imaging of GABA transients poses additional challenges due to overlapping biological and instrumental noise sources, making detection at single boutons more difficult than in quiescent brain slices. However, GABA released from interneuron bursts can diffuse tens of microns from its release sites, generating extracellular 'volume-transmitted' signals (*Oláh et al., 2009*). These GABA waves have previously been detected with iGABASnFR2 in brain slices during highly synchronized, pathological network discharges (*Magloire et al., 2023*). Here, we tested whether iGABASnFR2 could resolve volume-transmitted GABA release in response to physiological sensory stimulation in vivo.

We focused our imaging on areas of thalamocortical input in the barrel cortex L1-L3 and applied rhythmic whisker stimulation (RWS), a paradigm known to evoke reliable Ca²⁺ signals in thalamocortical axons (*Petreanu et al., 2012*), also shown in our own prior work (*Henneberger et al., 2020*).

Following expression of AAV9-hSyn-iGABASnFR2 in cortical neurons, a brief RWS protocol (four 20 Hz air puffs over 200 ms) consistently triggered robust fluorescence increases across a ~150 µm-wide area in the barrel cortex layers L2-L3 (see *Figure 6c*, or L1-L2 layers, *Figure 6—video 1*). Fluorescence changes were clearly visible across sensor-expressing neuronal processes, consistent with the interpretation that these are volume-transmitted signals generated by interneurons. Based on prior calibration of iGABASnFR2 in brain slices (*Magloire et al., 2023*), the observed $\Delta F/F_0$ corresponds to a transient extracellular GABA concentration increase of approximately 2–2.5 µM at peak.

## Discussion

Here we report an improved GABASnFR, developed using a high-throughput mutagenesis and screening pipeline that has previously been used to optimize calcium indicators (*Dana et al., 2016*; *Wardill et al., 2013*; *Zhang et al., 2023*). Despite the fact that only 20% of primary cultured neurons are expected to be inhibitory (*Wonders and Anderson, 2006*), this pipeline was still capable of identifying improved sensor variants. iGABASnFR2 has a sevenfold increase in affinity for GABA, as well as a 30% increase in rise time kinetics compared to the first-generation sensor. Importantly, sensor affinity remains in a range where it is not likely to be saturated by tonic levels of GABA. Additionally, the improved kinetics make it more likely the sensor can capture the rapid changes in GABA concentration that occur at the synaptic cleft as GABA is released and rapidly reuptaken.

The throughput of the screening platform made it possible to target 39 different sites in the protein for saturation mutagenesis. Such an extensive screen was important to get enough coverage that many sites with relatively small improvements could be combined to derive an overall more effective sensor. Of the 39 sites targeted, we found 12 mutations across 6 sites that contributed to improved sensor performance or expression. However, no single mutation produced an especially large improvement in performance. It was particularly challenging to identify single-site variants with both strong expression and high signal-to-noise. Fortunately, many of the single-site mutations interacted additively when combined. However, it was also the case that many double mutant combinations failed to pass our basic quality control standards, which may indicate that the combinations resulted in a failed, non-fluorescent sensor. Whether higher-order combinations can realistically yield sensors with improved performance remains an open question—functional variants may prove even more rare when more mutations are combined, and/or the increase in performance may come in smaller increments as higher-order combinations are produced.

No single mutation produced an especially large improvement in performance. Identifying single-site variants with both strong expression and high signal-to-noise was particularly challenging. Many of the single-site mutations interacted additively when combined. Still, numerous double mutant combinations failed to pass basic quality control standards, possibly indicating that the combinations produced non-functional, non-fluorescent sensors. Whether higher-order combinations can realistically yield sensors with improved performance remains an open question—functional variants may become even rarer as more mutations are added, and any performance gains may diminish with each additional layer of complexity.

The extensive screening also enabled the serendipitous discovery of negative-going sensors. Of course, positive- and negative-going variants cannot be combined in a single experiment because they have identical spectral properties. However, in experiments where precisely timed GABA release is expected, replicating an observation with sensors of different polarities could increase confidence in a result, particularly given the lower signal-to-noise of current GABA indicators compared to widely used sensors, such as GCaMP and iGluSnFR. More importantly, negative-going sensors convert decreases in extracellular GABA into positive fluorescence signals. Decreases in extrasynaptic GABA are a hallmark of changes in excitation–inhibition balance that accompany several important brain-state transitions. These include arousal and attention, where neuromodulatory drive increases GABA uptake and suppresses tonic interneuron and astrocytic GABA release, as well as stress- or antidepressant-responsive states linked to reduced astrocytic GABA production and increased transporter activity (*Semyanov et al., 2004*; *Farrant and Nusser, 2005*; *Brickley and Mody, 2012*; *Rusakov et al., 2011*; *Yoon et al., 2014*; *Ferguson and Gao, 2018*). In these contexts, a negative-going sensor offers a distinct advantage by providing a direct optical readout of shifts toward heightened network excitability that are difficult to infer from synaptic measurements alone.

There are several promising avenues that could be taken to further optimize iGABASnFR. First, although several sites contributed small improvements in sensor performance, we did not attempt to go much beyond two-site combinations in this study, so higher-order combinations are one straight-forward route to further optimization. Second, sensor expression is evidently an aspect which could be improved. We attempted to capture this in our screening criteria by measuring the fraction of respon-sive pixels for each variant. However, this partly conflates the amplitude of the sensor's signal with its expression level, and a more direct measure of expression would be preferable. Third, trafficking the sensor efficiently to the membrane is likely another point of improvement. In work with iGluSnFR (*Aggarwal et al., 2023*) and eLACCO (*Nasu et al., 2023*), a panel of different transmembrane anchors were evaluated for sensor localization, and some were found to be more effective than the PDGFR domain that anchors iGABASnFR2. Finally, the high throughput of our screening pipeline may provide sufficient training data that machine learning approaches used to further optimize jGCaMP8 (*Wait et al., 2024*) can be applied to iGABASnFR.

When assessing sensor performance, we observed consistent discrepancies between in vitro and cell-based measurements, with purified protein typically exhibiting higher apparent affinity and larger dynamic range than neuronal measurements. These discrepancies likely reflect both the addition of the ~60-amino-acid transmembrane anchor, which can constrain conformational flexibility, and the distinct physicochemical environment experienced by a membrane-tethered sensor at the neuronal surface. The magnitude and even direction of these effects are difficult to predict a priori, as illustrated by iGluSnFR3, which exhibits a higher apparent affinity when membrane-tethered than in soluble form (*Aggarwal et al., 2023*). Accordingly, we view biochemical characterization as an important guide during sensor optimization, but consider neuronal measurements the most informative indicator of in vivo sensor performance.

Importantly, even within these constraints, the improvements we identified resulted in a qualita-tive advance in the ability to detect inhibition in the retina. While the first-generation sensor showed some signal, direction-selective inhibition was reliably detectable on single trials only with iGAB-ASnFR2. This sensor has also been used to detect inhibitory transmission during epileptiform activity in hippocampal slices (*Magloire et al., 2023*). These are strong signals, but here we further show that iGABASnFR2 is sensitive enough to detect GABA release triggered by activation of single neurons in brain slices—a critical benchmark for in vivo applicability. In intact animals, the sensor reported GABA release in somatosensory cortex in response to physiological sensory stimulation. These signals reflect relatively slow volume-transmitted extracellular GABA 'waves' that result from spatiotemporal summation and dissipation of multiple GABA hotspots generated by rapid GABA release at individual synapses (*Boddum et al., 2016*; *Oláh et al., 2009*; *Pavlov et al., 2014*; *Sylantyev et al., 2020*). Although we did not distinguish synaptic from extrasynaptic GABA in this study, further improve-ments in kinetics and signal-to-noise may be needed to reliably resolve synaptic events and broaden the utility of the sensor for more demanding in vivo applications. Even in its current form, however, iGABASnFR2 is well suited for contexts where GABA signals are spatially integrated or temporally averaged, such as photometry or strong population-level activity. Given the longstanding difficulty of directly visualizing inhibitory signaling in the brain, we anticipate that these improved sensors will serve as valuable tools for the neuroscience community.

## Methods

### Animal care and use statement

All surgical and experimental procedures involving animals were performed in accordance with proto-cols approved by the Institutional Animal Care and Use Committee at the respective institute (HHMI Janelia Research Campus, National Institute of Genetics). Procedures in the United States conform to the National Institutes of Health (NIH) Guide for the Care and Use of Laboratory Animals. All animal procedures in the UK were conducted in accordance with the European Commission Directive (86/609/EEC), the United Kingdom Home Office (Scientific Procedures) Act (1986), with project approval from the Institutional Animal Care and Use Committees of the University College London. Procedures in Japan were approved by the animal experimentation committee at the National Institute of Genetics. Mice were housed under controlled temperature (roughly 21 °C) and humidity (roughly 50%) condi-tions under a reverse light cycle. This research has complied with all applicable ethical regulations.

## Mutagenesis

Site-directed mutagenesis was carried out using overlapping PCR amplicons with degenerate oligo-nucleotide primers (NNS). A single-site NNS library (n=96) was created for each amino acid residue of interest and cloned using Gibson assembly. Subsequently, top single-site variants (12 mutations across six AA sites) were assembled using overlapping amplicons to achieve all possible combinations of mutations (n=635 sequence-verified variants). For neuronal screening, iGABASnFR variants were cloned into an expression vector containing the CAG promoter, the variant of interest, and finally a woodchuck hepatitis virus post-transcriptional regulatory element (WPRE).

## Neuronal cell culture

The primary rat culture procedure was performed largely as previously described (*Dana et al., 2016*; *Wardill et al., 2013*). Briefly, neonatal rat pups (Charles River Laboratory) were euthanized and neocortices were dissociated and processed to form a cell pellet. Cells were resuspended and transfected by combining $5 \times 10^5$ viable cells with 400 ng plasmid DNA and nucleofection solution in a 25 µL electroporation cuvette (Lonza). Electroporation of iGABASnFR mutants was performed according to the manufacturer's protocol. Typically, eight wells in a 96-well plate were electroporated with iGABASnFR1 control, and 18 different mutants were electroporated into four wells each for a total of 80 wells. The first and last columns of the plate were not used because pilot experiments indicated that cells grown in these wells exhibited poor health.

## Field stimulation assay

The field stimulation assay was adapted from our existing GECI screening pipeline (*Dana et al., 2019*; *Dana et al., 2016*). Neurons were plated onto poly-D-lysine-coated, 96-well, glass bottom plates (MatTek) at ~$1 \times 10^5$ cells per well in 100 µL of a 1:4 mixture of NbActiv4 (BrainBits) and plating medium (28 mM glucose, 2.4 mM NaHCO3, 100 µg/mL transferrin, 25 µg/mL insulin, 2 mM L-glutamine, 100 U/mL penicillin, 10 µg/mL streptomycin, 10% FBS in MEM), and placed in an incubator at 37 °C and 5% $CO_2$. On the following day, 190 µL of NbActiv4 medium was added to each well, and incubation continued for a total of 12–15 days.

For imaging, neuronal culture growth medium was exchanged three times with imaging buffer (145 mM NaCl, 2.5 mM KCl, 10 mM glucose, 10 mM HEPES pH 7.4, 2 mM CaCl2, 1 mM MgCl2) and imaged in 75 µL of imaging buffer and drugs (10 µM CNQX, 10 µM (R)-CPP, 10 µM gabazine, 1 mM (S)-MCPG, Tocris). Fluorescence time-lapse images were collected on an Olympus IX81 microscope using a 10x 0.4 NA objective (UPlanSApo, Olympus) and an ET-GFP filter cube (Chroma #49002). Illumination was provided by a 470 nm LED (Cairn Research) and fluorescence was collected by an EMCCD camera (Ixon Ultra DU897, Andor) with 256×256 pixel frames taken at 50 Hz, with 2 x binning, over a 0.8 mm×0.8 mm FOV. A custom platinum electrode was lowered into each well via a linear actuator and neurons were stimulated (1, 10, 40 pulses at 83 Hz, 40 V, 1 ms pulse duration; S-48, Grass Instruments) as time-lapse imaging was performed.

The time-lapse images were analyzed to quantify the responses of the mutants to electrical stimulation. Photobleaching was corrected by fitting a single exponential to the beginning and end of the recording (avoiding the response window). For each pixel in the movie, a Mann-Whitney U test was performed between 1.5 s preceding the field stimulation and 0.4 s afterwards. Pixels with a *p*-value<0.01 for any of those tests were considered to be responsive. The number of responsive pixels was used as a proxy for sensor expression. In the screening phase, measurements of sensitivity (ΔF/F) and kinetics were performed on pooled responsive pixels from each plate, and wells with <4 responsive pixels were discarded. To account for batch variability, the response of each variant was normalized to the in-plate iGABASnFR1 control.

Quality control on every plate was performed initially by evaluating culture conditions and manually examining images from each well. General criteria for retesting were: aberrantly low expression, significant clumps of unhealthy cells or sparse cell distribution, out-of-range media pH, or data acquisition failures in the automated steps for focusing and field stimulus delivery. Individual variants with fewer than three replicates meeting quality criteria were flagged for retesting. If control iGABASnFR1 wells were flagged, the entire plate was rejected for potential systemic issues and all variants were retested. Overall, 54 plates did not pass this initial quality control step and were re-screened. Subsequently, all

variants with <1800 responsive pixels per plate (representing 0.6% of all pixels) were eliminated from analysis.

## Protein expression and purification

iGABASnFR variants of interest were expressed in *E. coli* T7 Express cells (New England BioLabs) in 100 mL cultures, pelleted by centrifugation, and frozen overnight. Cells were lysed by resuspending in buffer containing 20% v/v B-PER (ThermoFisher) and 0.2 mg/mL lysozyme. Lysate was sonicated, clarified by centrifugation, and purified using immobilized metal affinity chromatography on 1 mL nickel-charged HisPur resin (Thermo Fisher). The protein was eluted in a buffer containing 100 mM imidazole and concentrated by spin concentrator (Amicon).

## Crystal structure determination

All crystallization trials were carried out at room temperature using the hanging-drop vapor diffusion method. Commercial sparse-matrix screening solutions were used in initial screens (Hampton Research). 1 µL of protein solution containing 10 mg/mL of iGABASnFR2, 2 mM GABA, 20 mM Tris, 136 mM NaCl, pH 7.4 was mixed with 1 µL of reservoir solution and equilibrated against 250 µL reservoir solution. The reservoir solution for the optimized conditions contained 0.2 M ammonium formate and 20% w/v polyethylene glycol 3350. Diffraction data were collected at the beamline 8.2.1 at Berkeley Center for Structural Biology and processed with XDS (*Kabsch, 2010*). The phase was determined by molecular replacement using MOLREP and the structure of unliganded iGluSnFR precursor (PDB ID: 6DGV) without residue 1–98, 230–248, and 358–376 as the starting model. Subsequently, the model was rebuilt with Buccaneer (*Cowtan, 2006*). Refinement was performed using REFMAC, followed by manual remodeling with Coot (*Emsley et al., 2010*). The structure of the iGABASnFR2–GABA complex has been deposited in the Protein Data Bank (PDB ID: 9D57). Details of the crystallographic analysis and statistics are presented in *Figure 3—source data 1*.

## Affinity measurements

Equilibrium binding affinities were determined by titration with serial dilutions of GABA up to 10 mM into 0.2 µM protein solution in PBS. Measurements were made on a Tecan Spark plate reader using 490 nm excitation and 515 nm excitation at room temperature (22°C). The fluorescence data was fitted to the Hill function in GraphPad Prism to determine affinity values.

Competition assays (*Figure 4—figure supplement 2*) were performed similarly, except that GABA was titrated in the presence of 1 mM of a potential competing ligand (various amino acids or histamine) and compared with a control (PBS).

## pH titrations

Fluorescence intensities as a function of pH were measured in both glutamate-saturated (10 mM) and glutamate-free states, and fitted with a sigmoidal binding function to determine the apparent pKa and apparent ΔF/F as a function of pH. The sigmoidal binding function used was fit using Graphpad Prism v.10 software: $F=Fmin+(Fmax-Fmin)/(1+10^{(pKa-pH)})$, where F is the observed brightness, the parameters Fmax and Fmin are limit brightnesses at high and low pH, and pKa is the apparent pKa of the indicator.

## On-cell affinity measurements in neuronal culture

To measure the $EC_{50}$ of different sensor variants in culture conditions, neurons were transfected and plated in glass-bottom 24-well plates (Mattek) as described above. After seven days in culture, they were imaged using a 20x 0.45 NA objective (LUCPlanFL N, Olympus). GABA was perfused into each well using a custom 3D-printed part that held an inlet and outlet perfusion tube. The inlet tube was connected to a 4-channel gravity perfusion system (VC3, ALA Scientific), and the outlet tube was connected to a vacuum trap. Increasing concentrations of GABA from 0.1 µM to 10 mM were perfused into the culture for 5 s, followed by 9 s of GABA-free buffer, as imaging was performed.

## Stopped-flow kinetic measurements

GABA binding kinetics were measured using an Applied Photophysics SX20 stopped-flow spectrometer by rapidly mixing equal volumes of dilute purified sensor protein with varying GABA concentrations

in PBS buffer. Fluorescence changes were detected using 490 nm excitation and a 515 nm long-pass filter. Measurements were taken at room temperature (22°C). Three replicate measurements were performed for each set of conditions, and reaction traces were fitted to a monoexponential function in Matlab.

## One-photon photophysical measurements

All measurements were performed on solutions of purified protein (1–5 µM) in 1 X PBS (pH 7.3) with 0 mM (apo) and 10 mM (sat) added GABA. Absorbance measurements were performed using UV-Vis spectrometer (Cary 100, Agilent Technologies) and fluorescence excitation-emission spectra were measured using a fluorescence spectrometer (Cary Eclipse, Varian Inc). Excitation scans ranged from 250 to 530 nm and emission scans were performed from 470 to 700 nm. ΔF/F, defined as the difference in fluorescence from proteins +/-10 mM GABA, was calculated from the fluorescence emission spectra. Quantum yield measurements were performed via the relative method, using fluorescein dye (QY = 0.90 in 0.1 M NaOH) as a reference. Extinction coefficient measurements were determined using the alkali denaturation method, using the extinction coefficient of denatured GFP as a reference ($\varepsilon$=44,000 M-1 cm-1 at 447 nm).

## Two-photon measurements

Two-photon excitation spectra were measured as previously described (*Akerboom et al., 2012*). Protein solutions (1–5 µM)+/-10 mM GABA were prepared and measured using an inverted microscope (IX81, Olympus) equipped with a 60x 1.2 NA water immersion objective (Olympus). Two-photon excitation was achieved using an 80 MHz Ti-Sapphire laser (Chameleon Ultra II, Coherent) for spectra from 710 nm to 1080 nm. Fluorescence collected through the objective was passed through a short pass filter (720SP, Semrock) and a band pass filter (550BP200, Semrock), before being detected by a fiber-coupled Avalanche Photodiode (APD) (SPCM_AQRH-14, Perkin Elmer). The obtained two-photon excitation spectra were normalized for 1 µM concentration and further used to obtain action cross-section spectra (AXS) with fluorescein as a reference (Average AXS from *Makarov et al., 2008*; *Xu and Webb, 1996*).

## Imaging direction-selective activity in the retina

We injected AAVs encoding iGABASnFR1 (ssAAV-1/2-hSyn-FLEX.iGABASnFR.F102G) and iGAB-ASnFR2.0 (ssAAV-9/2-hSyn1-dlox-iGABASnFR2(rev)-dlox-WPRE-SV40p(A)) into ChAT-IRES-Cre mice to monitor the activity of starburst amacrine cells. Six to eight weeks after injection, we dissected retinae and performed two-photon imaging.

We used two types of visual stimuli: (1) Static flash: a spot with 500 µm diameter, 100% contrast, 2 s duration (*Figure 2d and g*); (2) Moving spot, a spot with 500 µm diameter, 800 µm/s (*Figure 2e and h*).

To quantify the response amplitude, we computed the response amplitude index (RAI):

$$RAI = \left(r_s - r_b\right) / \left(r_s + r_b\right)$$

where $r_s$ denotes the peak ΔF/F0 during visual stimulation and $r_b$ is the mean ΔF/F0 before stimulation.

To evaluate the response reliability, we examine the trial-to-trial variance of the signals (*Baden et al., 2016*),

$$Var[\langle C\rangle_r]_t / \langle Var[C]_t\rangle_r$$

where $C$ is a matrix constructed of response ΔF(t) in all stimulus trials, and $\langle\rangle_x$ and $Var$ denote the mean and variance across the indicated dimension $x$. If all responses are identical across all stimulus trials, reliability is equal to 1.

The signal-to-noise ratio (SNR) of motion responses was quantified as the response strength relative to the variance in baseline,

$$SNR = \left(R_p - SD_b\right) / \left(R_p + SD_b\right)$$

where $R_p$ denotes response amplitude in the preferred direction and $SD_b$ is the standard deviation of the signal before stimulation.

The circular variance (CV) is computed to quantify directional tuning:

$$CV = \left| \left( \Sigma_d R_d e^{i2\theta_d} \right) / \left( \Sigma_d R_d \right) \right|$$

where $R_d$ denotes response in direction $d$, and $\theta_d$ denotes angle in direction $d$.

## Viral transduction of iGABASnFR for acute slice experiments

We used viral transduction of the optical GABA sensor in C57BL/6 J mice (Charles River Laboratories), as detailed earlier (*Henneberger et al., 2020*; *Magloire et al., 2023*). Briefly, an AAV virus expressing iGABASnFR (Janelia Research Campus) was injected into the hippocampus during aseptic surgery. The animal, secured in a stereotaxic frame and under a stable anaesthesia level (maintenance at 1.5–2.5% isoflurane, body temperature maintained at ~37.0°C), received perioperative analgesics (subcutaneous buprenorphine, 60 µg kg⁻¹), and a craniotomy of approximately 1 mm diameter was performed using a high-speed drill. Stereotactic coordinates were 2.5 mm of the anteroposterior distance from bregma to lambda and 2.5 mm lateral to the midline. Pressure injections of the sensor (totalling 0.1E10 genomic copies in a volume not exceeding 200 nL) were stereotactically guided to a depth of 2.6 mm ventral from the cortical surface, at a rate of approximately 1 nL s⁻¹. Metacam (1 mg kg⁻¹) and saline (0.5 ml) were subcutaneously administered.

Acute hippocampal slices (350 µm thick) were prepared from mice 3–5 weeks after in vivo transduction. Brains were rapidly extracted and sliced in ice-cold cutting solution containing (in mM): 75 NaCl, 80 sucrose, 2.5 KCl, 7 MgCl$_2$, 1.25 NaH$_2$PO$_4$, 0.5 CaCl$_2$, 26 NaHCO$_3$, 1.3 ascorbic acid, 3 sodium pyruvate, and 6 glucose (osmolarity 300–305 mOsm). Slices were incubated at 34 °C in the same solution for 15 min, then transferred to an interface chamber containing artificial cerebrospinal fluid (aCSF) composed of (in mM): 126 NaCl, 26 NaHCO$_3$, 2.5 KCl, 2 CaCl$_2$, 1.3 MgSO$_4$, 1 NaH$_2$PO$_4$, and 10 glucose (pH 7.4, 295–305 mOsm). All solutions were continuously bubbled with 95% O$_2$ and 5% CO$_2$. Slices were allowed to recover for at least 60 min prior to recording. During experiments, slices were transferred to a submersion-type recording chamber and superfused with extracellular solution at 33–35 °C.

## Organotypic slice preparation

Organotypic hippocampal slice cultures were prepared from postnatal day 6–8 (P6–8) mice using a modified version of the interface culture method (*Stoppini et al., 1991*). Hippocampi were sectioned at 300 µm using a Leica VT1200S vibratome in ice-cold sterile slicing solution containing (in mM): 105 sucrose, 50 NaCl, 2.5 KCl, 1.25 NaH$_2$PO$_4$, 7 MgCl$_2$, 0.5 CaCl$_2$, 1.3 ascorbic acid, 3 sodium pyruvate, 26 NaHCO$_3$, and 10 glucose. Slices were washed in culture medium composed of 50% Minimal Essential Medium, 25% horse serum, 25% Hanks' Balanced Salt Solution, 0.5% L-glutamine, 28 mM glucose, and antibiotics (100 U/mL penicillin and 100 µg/mL streptomycin). Three to four slices were placed on each 0.4 µm pore membrane insert (Millicell-CM, Millipore, UK) and maintained at 37 °C in a humidified 5% CO$_2$ incubator. Medium was exchanged every 2–3 days, and cultures were maintained for up to 21 days in vitro (DIV). For electrophysiological recordings, slices were transferred to a submersion-type chamber and perfused with oxygenated aCSF. Recordings were performed at 33–35 °C. To minimize plasticity-related changes during prolonged recordings, the aCSF was supplemented with 10 µM NBQX and 50 µM AP5.

## Biolistic transfection of iGABASnFR variants

iGABASnFR2 was expressed under the synapsin promoter in CA3 interneurons in organotypic hippocampal slice cultures using biolistic transfection techniques adapted from manufacturer's instructions (Bio-Rad), as detailed earlier (*Jensen et al., 2017*). Gold micro-carriers (1.6 µm; 6–8 mg) were coated with 30 µg of plasmid DNA. At 5 days in vitro (DIV), cultures were treated overnight with culture media containing 5 µM Ara-C to reduce glial reaction following transfection. The following day, slices were transfected using a Helios gene gun (Bio-Rad) at 120 psi. Cultures were then returned to standard media and maintained for an additional 5–10 days before recordings.

## Electrophysiology and axonal imaging in hippocampal slices

Whole-cell patch-clamp recordings were obtained from visually identified CA1 pyramidal neurons using infrared differential interference contrast (IR-DIC) microscopy. Recording electrodes were fabricated from thin-walled borosilicate glass capillaries and had resistances of 2.5–3.5 MΩ. The internal pipette solution contained (in mM): 135 $KCH_3O_3S$, 10 HEPES, 10 $Na_2$-phosphocreatine or di-Tris-phosphocreatine, 4 $MgCl_2$, 4 $Na_2$-ATP, 0.4 Na-GTP, and 5 QX-314 bromide. The pH was adjusted to 7.2 using KOH, and osmolarity was maintained at 290–295 mOsm. Alexa Fluor 594 hydrazide (50 µM) was included to visualize cell morphology and identify synaptic boutons.

Recordings were performed in voltage-clamp mode with cells held at −65 mV. The extracellular aCSF was supplemented with 100 µM picrotoxin (Tocris Bioscience) and 30 µM D-serine. Electrophysiological signals were acquired using a Multiclamp 700B amplifier (Molecular Devices) and a motorized micromanipulator and XY translation stage system (Luigs and Neumann). Signals were digitized at 10 kHz and stored for offline analysis using either WinWCP (versions 4.1.5–4.7; John Dempster, University of Strathclyde) or pCLAMP (versions 10.4–10.5; Molecular Devices).

To visualize and trace axons of patched interneurons, we used a Femtonics Femto2D (or 3D)-FLIM imaging system integrated with patch-clamp electrophysiology (Femtonics, Budapest) and linked on the same light path to two femtosecond pulsed lasers MaiTai (SpectraPhysics-Newport) with independent shutter and intensity control. For CA3 pyramidal cells, the internal solution contained (in mM) 135 potassium methane-sulfonate, 10 HEPES, 10 di-Tris-Phosphocreatine, 4 $MgCl_2$, 4 $Na_2$-ATP, and 0.4 Na-GTP (pH adjusted to 7.2 using KOH, osmolarity 290–295). Presynaptic imaging was carried out using an adaptation of previously described methods for separate presynaptic glutamate and Ca2+ imaging (*Jensen et al., 2017*).

iGABASnFR-expressing cells were identified using two-photon imaging at 910 nm and patched in whole-cell configuration using a 25x Olympus objective (NA 1.05) for visualization. After the break-in, 30–45 min was allowed for Alexa Fluor 594 to equilibrate throughout the axonal arbor. Axons, identified by their smooth morphology and often tortuous trajectory, were traced to their targets using frame scan mode. If iGABASnFR fluorescence was too weak, we switched to the Alexa channel to trace the axon using 800 nm excitation. Discrete boutons were identified using established morphological criteria. Tornado scans of 1.5 micron diameter were set over the pre-synaptic bouton and scanned at a rate of 500 Hz with a train of action potentials initiated by brief positive voltage steps. While the laser power was reduced to minimize photobleaching, any detectable photobleaching trends were corrected in individual recordings a posteriori.

## Viral transduction of synGABASnFR2 in the barrel cortex

Male and female C57BL/6 N mice (1–1.5 months old) were prepared for aseptic surgery. Perioperative multimodal analgesia utilized buprenorphine (60 µg kg-1, s.c.) and lidocaine (2.5%) topically applied to the surgical site; ocular ointment (Lacri-lube, Allergan, UK) was applied to prevent corneal drying. Anesthesia was induced with 5% isoflurane and maintained at 1.5–2.5%. After shaving, the scalp was disinfected with three washes of chlorhexidine, and the mouse was secured in a stereotaxic frame (David Kopf Instruments, USA). Adequate anesthesia was confirmed by loss of pedal reflexes, and body temperature was maintained at ~37 °C using a feedback-controlled heating system.

A midline incision was made and overlying soft tissue resected to expose the skull. A small craniotomy (~1 mm diameter) was drilled using a high-speed dental drill at coordinates targeting either the somatosensory cortex (AP: −1.5 mm, ML: −3.0 mm relative to bregma) or hippocampus (AP: −1.9 mm, ML: −1.4 mm, DV: −1.35 mm). AAV9-hSyn-GABASnFR2 was injected via a Hamilton syringe stereotactically positioned 1 mm below the cortical surface. Injections (300–500 nL total volume) were delivered at ~20 nL/min using a microinjection pump in three steps, retracting the needle by 150–200 µm after each step. After completion, the needle was left in place for ~5 min before withdrawal. The wound was sutured, and animals received meloxicam (1 mg/kg) and 0.5 mL saline. Recovery took place in a heated chamber, with postoperative monitoring until full wound healing (typically 2–4 days).

## Cranial window implantation

Mice were anesthetized and positioned in a stereotaxic frame as described for the viral transduction procedure. Once stabilized under general anesthesia (isoflurane, maintained at 1.5–2%), a large

portion of the scalp was removed to expose the right frontal and parietal skull bones, along with the medial regions of the left frontal and parietal bones. The right temporalis muscle was reflected laterally to expose the squamous suture, to facilitate cement bonding during fixation of the cranial window implant. The exposed skull surface was coated with Vetbond (3 M, USA), and a custom-designed headplate was mounted over the S1BF using dental cement (SuperBond, Sun Medical Co. Ltd., Japan).

After the bonding agents had fully cured, the animal was transferred to a custom-built head fixation setup. A 3 mm craniotomy was made over the right somatosensory cortex, centered over the S1BF injection site, using a high-speed hand drill. After sufficiently thinning the skull, prior to removal of the skull flap, the surface was superfused with warmed aCSF (in mM: 125 NaCl, 2.5 KCl, 26 NaHCO$_3$, 1.25 Na$_2$HPO$_4$, 10 glucose, 2 CaCl$_2$, 2 MgSO$_4$ saturated with 95% O$_2$/5% CO$_2$ pH 7.4). The skull flap was removed using fine-tipped forceps (11252–40, Fine Science Tools, Germany). Immediately after opening, the brain was superfused with sterile saline. The durotomy was performed using 26 G needles with hand-made curved tips (modified by tapping against a hard surface), ensuring the pia mater remained intact.

A custom-prepared cranial window—a double-glass assembly consisting of a 34 mm diameter round coverglass bonded beneath a 4 mm diameter round coverglass (Harvard Apparatus UK), affixed with UV-curable optical adhesive (NOA61, ThorLabs Inc, NJ, USA)—was placed over the exposed cortex. Slight downward pressure was applied to the coverslip using a stereotactically guided wooden spatula to prevent excessive force to the open brain surface, as described previously (*Henneberger et al., 2020*). Superfusion was stopped, and excess aCSF was carefully absorbed with a sterile surgical sponge, ensuring no fluid remained beneath the cranial window. The coverslip was secured with dental cement. After completing the surgery, the anaesthesia regime was switched from inhalation to i.p. injection, using a mixture of fentanyl (0.03 mg/kg), midazolam (3 mg/kg), and medetomidine (0.3 mg/kg), for subsequent imaging in the anaesthetised animal.

## Multiplexed two-photon imaging in vivo

The animal was transferred to the Femtosmart imaging system (Femtonics, Budapest), integrated with electrophysiology and connected to a tunable femtosecond pulsed laser (Insight X3, Spectra-Physics/Newport) for multiplexed two-photon excitation (2PE) imaging. Under anesthesia, the animal was secured on a custom-built stage via the implanted headplate, beneath an XLPlan N 25x water-immersion objective (NA 1.05) and illuminated with a green lamp.

2PE imaging was performed at 910 nm, optimized for the sensor. Imaging was conducted in cortical layers 1 and 2/3 at depths ranging from 50 to 300 µm, adjusting laser intensity to minimize photobleaching (<15 mW). Once a suitable region was identified, data were acquired using either Tornado (spiral) linescan mode (0.5–1 kHz) or resonant frame scan mode. Frame scan time series were typically recorded as 512×50×317 pixels at 200–300 Hz for 5 s. Recordings were acquired using MESc v3.5.7 (Femtonics) and exported as t-stacks for further processing in ImageJ. In Tornado scan mode, circular fields of view (50–150 µm in diameter) were selected to encompass iGABASnFR2-positive cells. Given that GABA released from brief interneuronal bursts is expected to diffuse across extracellular distances on the order of 10 µm, we opted to image relatively broad imaging fields of view (50–150 µm) rather than targeting individual presynaptic boutons.

## Statistics and reproducibility

Values and errors reported throughout the text are mean ± s.e.m. unless otherwise noted. Statistical analyses were performed in GraphPad Prism v.8 (in vitro photophysics), MATLAB, Python, and OriginPro, as described in the text.

## Reagent availability

A variety of mammalian and bacterial expression plasmids as well as AAVs for iGABASnFR2 and iGABASnFR2n are available through Addgene (https://www.addgene.org/browse/article/28247749/).

Requests for other reagents, such as sensor variants with different affinities, can be made by contacting GENIEreagents@janelia.hhmi.org.

## Acknowledgements

We thank Tim Brown from the Janelia Tool Translation Team, Deepika Walpita, and Phuong Nguyen for providing resources. Portions of the manuscript text and analysis code were drafted and refined with assistance from large language models (ChatGPT, OpenAI; Claude, Anthropic), under the supervision of the authors. Work at Janelia research campus was funded by the Howard Hughes Medical Institute. AM was supported by funding from PRESTO JST (JPMJPR2489) and KAKENHI (23K19412). KY was supported by funding from KAKENHI (20K23377; 22K21353; 23H04241; 24H02311). LLL was supported by a Wellcome Collaborative Award (223131/Z/21/Z); DAR was supported by funding from MRC Research Grant (ref MR/W019752/1), Wellcome Collaborative Award (223131/Z/21/Z), and BBSRC Research Grant (PI; ref BB/Y003926/1). OK was supported by UCL-Wellcome Trust Translational Partnership Pilot Award (RM-TIN OK: #178973). DAR, OK, and TJ thank Dr. James Reynolds for help with setting up in vivo viral transduction protocols.

---

## Additional information

### Funding

| Funder | Grant reference number | Author |
| --- | --- | --- |
| PRESTO JST | 10.52926/jpmjpr2489 | Akihiro Matsumoto |
| KAKENHI | 20K23377 | Keisuke Yonehara |
| KAKENHI | 22K21353 | Keisuke Yonehara |
| KAKENHI | 23H04241 | Keisuke Yonehara |
| KAKENHI | 24H02311 | Keisuke Yonehara |
| Wellcome Trust | 10.35802/223131 | Loren L Looger<br>Dmitri A Rusakov |
| Medical Research Council | MR/W019752/1 | Dmitri A Rusakov |
| Biotechnology and Biological Sciences Research Council | BB/Y003926/1 | Dmitri A Rusakov |
| UCL-Wellcome Trust Translational Partnership Pilot Award | #178973 | Olga Kopach |
| Howard Hughes Medical Institute | | Ilya Kolb<br>Jeremy P Hasseman<br>Benjamin J Arthur<br>Yan Zhang<br>Arthur Tsang<br>Daniel Reep<br>Getahun Tsegaye<br>Jihong Zheng<br>Ronak Patel<br>Loren Looger<br>Jonathan S Marvin<br>Wyatt Korff<br>GENIE Project Team<br>Glenn C Turner |
| KAKENHI | 23K19412 | Akihiro Matsumoto |

The funders had no role in study design, data collection and interpretation, or the decision to submit the work for publication. For the purpose of Open Access, the authors have applied a CC BY public copyright license to any Author Accepted Manuscript version arising from this submission.

## Author contributions
Ilya Kolb, Conceptualization, Resources, Data curation, Software, Formal analysis, Supervision, Investigation, Visualization, Methodology, Writing – original draft, Project administration, Writing – review and editing; Jeremy P Hasseman, Conceptualization, Data curation, Supervision, Investigation, Methodology, Writing – original draft, Project administration, Writing – review and editing; Akihiro Matsumoto, Data curation, Formal analysis, Funding acquisition, Validation, Investigation, Methodology, Writing – review and editing; Thomas P Jensen, Olga Kopach, Data curation, Formal analysis, Funding acquisition, Investigation, Visualization, Methodology, Writing – review and editing; Benjamin J Arthur, Data curation, Software, Formal analysis, Visualization, Writing – review and editing; Yan Zhang, Data curation, Formal analysis, Investigation, Visualization, Methodology, Writing – original draft, Writing – review and editing; Arthur Tsang, Data curation, Investigation, Methodology, Writing – review and editing; Daniel Reep, Data curation, Investigation, Methodology; Getahun Tsegaye, Investigation, Methodology; Jihong Zheng, Data curation, Formal analysis, Investigation, Methodology; Ronak H Patel, Investigation, Methodology, Writing – review and editing; Loren L Looger, Conceptualization, Supervision, Project administration, Writing – review and editing; Jonathan S Marvin, Conceptualization, Supervision, Writing – original draft, Writing – review and editing; Wyatt L Korff, GENIE Project Team, Supervision, Project administration; Dmitri A Rusakov, Conceptualization, Data curation, Supervision, Funding acquisition, Visualization, Writing – original draft, Project administration, Writing – review and editing; Keisuke Yonehara, Conceptualization, Supervision, Funding acquisition, Visualization, Writing – original draft, Project administration, Writing – review and editing; Glenn C Turner, Data curation, Software, Formal analysis, Supervision, Investigation, Visualization, Writing – original draft, Project administration, Writing – review and editing

## Author ORCIDs
Ilya Kolb ⓘ https://orcid.org/0000-0001-5100-849X
Jeremy P Hasseman ⓘ https://orcid.org/0000-0002-0096-7321
Akihiro Matsumoto ⓘ https://orcid.org/0000-0002-8552-1416
Thomas P Jensen ⓘ https://orcid.org/0000-0002-2560-6784
Olga Kopach ⓘ https://orcid.org/0000-0002-3921-3674
Benjamin J Arthur ⓘ https://orcid.org/0000-0003-3545-8807
Yan Zhang ⓘ https://orcid.org/0000-0001-8376-3190
Arthur Tsang ⓘ https://orcid.org/0009-0005-5173-1211
Daniel Reep ⓘ https://orcid.org/0009-0009-7638-9538
Getahun Tsegaye ⓘ https://orcid.org/0009-0005-3665-0907
Jihong Zheng ⓘ https://orcid.org/0000-0002-1247-6087
Ronak H Patel ⓘ https://orcid.org/0000-0002-6945-6466
Loren L Looger ⓘ https://orcid.org/0000-0002-7531-1757
Jonathan S Marvin ⓘ https://orcid.org/0000-0003-2294-4515
Wyatt L Korff ⓘ https://orcid.org/0000-0001-8396-1533
Dmitri A Rusakov ⓘ https://orcid.org/0000-0001-9539-9947
Keisuke Yonehara ⓘ https://orcid.org/0000-0001-9541-6671
Glenn C Turner ⓘ https://orcid.org/0000-0002-5341-2784

## Ethics
All surgical and experimental procedures involving animals were performed in accordance with protocols approved by the Institutional Animal Care and Use Committee at the respective institute (HHMI Janelia Research Campus, National Institute of Genetics). Procedures in the United States conform to the National Institutes of Health (NIH) Guide for the Care and Use of Laboratory Animals. All animal procedures in the UK were conducted in accordance with the European Commission Directive (86/609/EEC), the United Kingdom Home Office (Scientific Procedures) Act (1986), with project approval from the Institutional Animal Care and Use Committees of the University College London. Procedures in Japan were approved by the animal experimentation committee at the National Institute of Genetics. Mice were housed under controlled temperature (roughly 21°C) and humidity (roughly 50%) conditions under a reverse light cycle. This research has complied with all applicable ethical regulations.

Reviewer #1 (Public review): https://doi.org/10.7554/eLife.108319.3.sa1
Author response https://doi.org/10.7554/eLife.108319.3.sa2

## Additional files

### Supplementary files
MDAR checklist

### Data availability
All analysis code and source data are available at https://doi.org/10.5281/zenodo.17971100. The crystal structure has been deposited in the Protein Data Bank under accession code 9D57 https://doi.org/10.2210/pdb9D57/pdb.

The following datasets were generated:

| Author(s) | Year | Dataset title | Dataset URL | Database and Identifier |
|---|---|---|---|---|
| Turner G | 2025 | iGABASnFR2: Improved genetically encoded protein sensors of GABA | https://zenodo.org/records/17971101 | Zenodo, 10.5281/zenodo.17971101 |
| Zhang Y, Looger LL | 2025 | iGABASnFR2 fluorescent GABA sensor in complex with GABA | https://www.wwpdb.org/pdb?id=pdb_00009d57 | Worldwide Protein Bank, 00009d57 |

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
