## [Editor Report · eLife Assessment]

This manuscript reports the development and characterization of iGABASnFR2, a genetically encoded GABA sensor that demonstrates substantially improved performance compared to its predecessor, iGABASnFR1. The work is comprehensive and methodologically rigorous, combining high-throughput mutagenesis, functional screening, structural analysis, biophysical characterization, and in vivo validation. The significance of the findings is **fundamental**, and the supporting evidence is **compelling**. iGABASnFR2 represents a notable advance in GABA sensor engineering, enabling enhanced imaging of GABA transmission both in brain slices and in vivo, and constitutes a timely, technically robust addition to the molecular toolkit for neuroscience research.

---

## [Referee Report · Reviewer #1 (Public review)]

Summary:

This manuscript by Kolb and Hasseman et al. introduces a significantly improved GABA sensor, building on the pioneering work of the Janelia team. Given GABA's role as the main inhibitory neurotransmitter and the historical lack of effective optical tools for real-time in vivo GABA dynamics, this development is particularly impactful. The new sensor boasts an enhanced signal-to-noise ratio (SNR) and appropriate kinetics for detecting GABA dynamics in both in vitro and in vivo settings. The study is well-presented, with convincing and high-quality data, making this tool a valuable asset for future research into GABAergic signaling.

Strengths:

The core strength of this work lies in its significant advancement of GABA sensing technology. The authors have successfully developed a sensor with higher SNR and suitable kinetics, enabling the detection of GABA dynamics both in vitro and in vivo. This addresses a critical gap in neuroscience research, offering a much-needed optical tool for understanding the most important inhibitory neurotransmitter. The clear representation of the work and the convincing, high-quality data further bolster the manuscript's strengths, indicating the sensor's reliability and potential utility. We anticipate this tool will be invaluable for further investigation of GABAergic signaling.

Weaknesses:

Despite the notable progress, a key limitation is that the current generation of GABA sensors, including the one presented here, still exhibits inferior performance compared to state-of-the-art glutamate sensors. While this work is a substantial leap forward, it highlights that further improvements in GABA sensor would still be highly beneficial for the field to match the capabilities seen with glutamate sensors.

---

## [Author Response]

The following is the authors’ response to the original reviews

**Public Reviews:**

**Reviewer #1 (Public review):**
Summary:This manuscript by Kolb and Hasseman et al. introduces a significantly improved GABA sensor, building on the pioneering work of the Janelia team. Given GABA's role as the main inhibitory neurotransmitter and the historical lack of effective optical tools for real-time in vivo GABA dynamics, this development is particularly impactful. The new sensor boasts an enhanced signal-to-noise ratio (SNR) and appropriate kinetics for detecting GABA dynamics in both in vitro and in vivo settings. The study is well-presented, with convincing and high-quality data, making this tool a valuable asset for future research into GABAergic signaling.Strengths:The core strength of this work lies in its significant advancement of GABA sensing technology. The authors have successfully developed a sensor with higher SNR and suitable kinetics, enabling the detection of GABA dynamics both in vitro and in vivo.This addresses a critical gap in neuroscience research, offering a much-needed optical tool for understanding the most important inhibitory neurotransmitter. The clear representation of the work and the convincing, high-quality data further bolster the manuscript's strengths, indicating the sensor's reliability and potential utility. We anticipate this tool will be invaluable for further investigation of GABAergic signaling.Weaknesses:Despite the notable progress, a key limitation is that the current generation of GABA sensors, including the one presented here, still exhibits inferior performance compared to state-of-the-art glutamate sensors. While this work is a substantial leap forward, it highlights that further improvements in GABA sensors would still be highly beneficial for the field to match the capabilities seen with glutamate sensors.

We thank Reviewer 1 for the positive assessment. We agree that further improvements in GABA sensor performance remain desirable. We acknowledge this limitation and outline directions for future development in the Discussion paragraph beginning "There are several promising avenues that could be taken to further optimize iGABASnFR."

**Reviewer #2 (Public review):**
Summary:This manuscript presents the development and characterization of iGABASnFR2, a genetically encoded GABA sensor with markedly improved performance over its predecessor, iGABASnFR1. The study is comprehensive and methodologically rigorous, integrating high-throughput mutagenesis, functional screening, structural analysis, biophysical characterization, and in vivo validation. iGABASnFR2 represents a significant advancement in GABA sensor engineering and application in imaging GABA transmission in slice and in vivo. This is a timely and technically strong contribution to the molecular toolkit for neuroscience.Strengths:The authors apply a well-established sensor optimization pipeline and iterative engineering strategy from single-site to combinatorial mutants to engineer iGABASnFR2. The development of both positive and negative going variants (iGABASnFR2 and iGABASnFR2n) offers experimental flexibility. The structure and interpretation of the key mutations provide insights into the working mechanism of the sensor, which also suggest optimization strategies. Although individual improvements in intrinsic properties are incremental, their combined effect yields clear functional gains, enabling detection of direction-selective GABA release in the retina and volume-transmitted GABA signaling in somatosensory cortex, which were challenging or missed using iGABASnFR1.Weaknesses:With minor revisions and clarifications, especially regarding membrane trafficking, this manuscript will be a valuable resource for probing inhibitory transmission.

We thank Reviewer 2 for the positive assessment. Regarding membrane trafficking, we appreciate the suggestion to test different trafficking motifs. While such optimization represents a valuable direction for future development, it was beyond the scope of the present study and not feasible with the available time and resources. A different imaging modality would be needed to assess membrane trafficking efficiency or membrane-restricted expression, as the images presented in the manuscript (Figure 2a) are wide-field epifluorescence images, which lack the axial resolution required to distinguish membrane-localized signal from cytosolic fluorescence.

We expect that the current characterization of iGABASnFR2 will nevertheless provide a strong foundation for future efforts to optimize membrane targeting and expression using alternative trafficking strategies.

**Recommendations for the authors:**

**Reviewer #1 (Recommendations for the authors):**
(1) We noted an interesting inconsistency in the response of iGABASnFR1 and iGABASnFR2 when expressed as purified protein versus in mammalian cells. Such discrepancies are not uncommon for proteins exhibiting different behaviors in *E. coli* versus mammalian expression systems. We appreciate the authors' diligent effort in performing screening within a neuronal context. Similarly, the stark difference between the absolute affinity in purified form (∼0.778 μM) and on-cell measurements (6.4 μM) warrants further discussion. The authors may consider commenting on these observations in the discussion section.

We have revised the Discussion (lines 401-410 in the ‘Tracked Changes’ document) to address the discrepancy between measurements obtained with purified protein and those from expression on the neuronal surface. As noted by the reviewer, such discrepancies are common, and our revision is intended to convey our empirical experience with this phenomenon rather than to offer a definitive mechanistic explanation.

One factor to appreciate is that, when on the surface of neurons, the sensor is tethered to the membrane by an additional 60 amino acids. In addition to altering the local chemical environment, membrane tethering could impose entropic or mechanical constraints on the sensor. These constraints may damp conformational motions that underlie ligand binding and fluorescence changes. Beyond this, the local environment experienced by a membrane-anchored sensor differs substantially from that of soluble protein. There are potential electrostatic and steric effects arising from the plasma membrane and extracellular matrix, as well as post-translational modifications associated with mammalian expression. These effects on sensor performance are not readily predictable in either magnitude or direction, as illustrated by iGluSnFR, which exhibits a higher apparent affinity when membrane-tethered than in soluble form (Aggarwal et al 2023). For these reasons, we place greater emphasis on neuronal measurements as the most informative indicator of in vivo sensor performance.

(2) Although iGABASnFR2 fluorescence exhibits pH dependence, its response appears less pH-dependent compared to the first-generation sensor. To enhance clarity, we suggest plotting the normalized response of both sensors across different pH values. This visual representation would be highly informative for readers.

Thank you - we have implemented this, now showing the (F_sat - F_apo)/F_apo response as a function of pH for all three sensors in Fig 4 fig. supp 3b. This visualization nicely illustrates that the apo-to-sat response of iGABASnFR1 is much more influenced by pH than either iGABASnFR2 or iGABASnFR2n, which we note on lines 252-253 of the ‘Tracked Changes’ document.

(3) To provide a more comprehensive characterization of the sensors, we recommend including a quantification of the decay times for all three versions of the sensors in Figure 2, specifically after panel 2c.

Thank you - we now provide this in Fig 2d.

(4) For improved readability of Figure 3a, we suggest adding distinct labels for iGABASnFR1 and iGABASnFR2 with corresponding colors.

Good suggestion - we matched the color of the backbones to the rest of the manuscript (orange and green). We also added labels on the figure to ensure clarity.

(5) The GABA released by SAC cells in Figure 5 looks amazing! We propose a minor modification to the cartoon in Figure 5b: mirroring the image horizontally (left to right). Given that the subsequent panels (e, h, and k) set the preferred direction of SAC movement as rightward, the current cartoon in Figure 5b inadvertently suggests stronger inhibition by SAC-released GABA when the spot moves left. Mirroring the image would align the cartoon more accurately with the subsequent data representations.

Thanks - this is a nice streamlining. We have implemented the change.

**Reviewer #2 (Recommendations for the authors):**
(1) As sensor performance differs substantially between purified protein and neurons, a summary table comparing key properties (e.g., EC50, ∆F/F _ax_, response amplitude to # of AP) across purified protein and neurons would be highly informative.

We discuss differences in sensor performance between purified protein and neurons in the Discussion (lines 401-410 in ‘Tracked Changes document) and, for the reasons outlined there, consider neuronal measurements to be far more predictive of in vivo performance. We therefore chose not to include a summary table directly comparing purified protein and neuronal data, as this would risk over-emphasizing in vitro measurements that we view primarily as qualitative signposts rather than more directly informative indicators of functional performance.

(2) The authors should comment on the observed differences in performance between purified protein and neuronal expression. Would HEK293 cell measurements serve as a better predictor of in vivo performance than in vitro titrations? Insights here would benefit future sensor development pipelines.

We have revised the Discussion to address this point (lines 401-410 in the ‘Tracked Changes’ document). We often observe differences in sensor performance between purified protein measurements and cellular or in vivo contexts. In our experience, titrations in primary neurons provide a better predictor of in vivo performance than in vitro protein titrations, as they more closely reflect relevant cellular factors. We do not have direct evidence that expression in heterologous systems such as HEK293 cells is generally more predictive, although this seems plausible; however, predictions inevitably become less reliable as sensors are translated to fully in vivo conditions.

(3) Improved membrane localization likely contributes to the enhanced sensitivity of iGABASnFR2 in neurons beyond changes in EC50. In Figure 2a, membrane trafficking appears suboptimal. The authors should explore alternative trafficking motifs (e.g., ER2, Kv2.1, or motifs from other sensors) to further improve the membrane expression and consider adding a second fluorescent protein for quantifying membrane-localized brightness.

Figure 2a presents wide-field epifluorescence images, which lack the axial resolution required to distinguish membrane-localized signal from cytosolic fluorescence. We therefore do not consider this imaging modality suitable for assessing membrane trafficking efficiency or membrane-restricted expression.

We appreciate the suggestion to test different trafficking motifs to attempt to better capture biological signals. While such optimization represents a valuable direction for future development, it was beyond the scope of the present study and not feasible with the available time and resources. We expect that the current characterization of iGABASnFR2 will nevertheless provide a strong foundation for future efforts to further optimize membrane targeting and expression using alternative trafficking strategies.

(4) Figure 4 - Supplement 2: The apparent EC50 of iGABASnFR2 seems affected by buffer composition and the presence of high concentrations of unrelated compounds. The authors should comment on this.

We thank the reviewer for raising this point. Upon closer inspection, the EC50 of iGABASnFR2 in Fig 4 Supp 2 is measured at 1.4 μM, while in Fig 4a it is 1.1 μM - these mean values are quite close to one another, and within the range of experimental variability we expect for experiments done weeks or months apart. What differs most noticeably in this dataset is the shape of the dose–response curve rather than the EC50 itself; the origin of this difference is currently unclear. We have revised the Results text (lines 226-231 in ‘Tracked Changes document) to clarify this point and to emphasize that the key observation of Fig. 4–figure supplement 2 is that none of the additional compounds tested substantially impair GABA binding, indicating that they do not act as strong non-competitive allosteric antagonists or inhibitors.

(5) The negative-going variant, iGABASnFR2n, is introduced but only briefly characterized. Including additional data or even a conceptual use case would clarify its potential utility.

We have modified the discussion to provide more examples of conceptual use cases, clarifying how such a sensor could indeed be highly impactful. The full passage is lines 372-387 in the ‘Tracked Changes’ document; to summarize: a key application of the negative-going sensor is detecting decreases in ‘GABA tone’, which plays a key role in setting the excitation-inhibition balance across brain circuits. Reductions in extrasynaptic GABA are a well-documented feature of several biologically important brain-state transitions, including arousal, experience-dependent plasticity, and stress-related modulation of inhibition, and iGABASnFR2n could be an important tool for investigating these processes.